# Teichoic acids in the periplasm and cell envelope of *Streptococcus pneumoniae*

**Mai Nguyen[1][†], Elda Bauda[1][†], Célia Boyat[1], Cédric Laguri[1], Céline Freton[1], Anne Chouquet[1], Benoit Gallet[1], Morgane Baudoin[2], Yung-Sing Wong[2], Christophe Grangeasse[3], Christine Moriscot[3], Claire Durmort[1], André Zapun[1]\*, Cecile Morlot[1]\***

[1]Univ. Grenoble Alpes, CNRS, CEA, IBS, Grenoble, France; [2]Univ. Grenoble Alpes, CNRS, DPM, Grenoble, France; [3]Molecular Microbiology and Structural Biochemistry, Université de Lyon, CNRS, Lyon, France

**\*For correspondence:**
andre.zapun@ibs.fr (AZ);
cecile.morlot@ibs.fr (CM)

[†]These authors contributed
equally to this work

**Competing interest:** The authors
declare that no competing
interests exist.

**Reviewing Editor:** Christopher
Ealand, The University of the
Witwatersrand, South Africa

## eLife Assessment

The bacterial cell wall is crucial to maintain viability. It has previously been suggested that Gram positive bacteria have a periplasmic region between the cell membrane and peptidoglycan cell wall that this is maintained by the presence of teichoic acids. In this **valuable** study, Nguyen et al. make clever use of electron microscopy and metabolic labelling to interrogate the role of teichoic acids in supporting the maintenance of the periplasmic region in Streptococcus pneumoniae. The findings are **solid** and close some crucial knowledge gaps whilst providing novel tools to further interrogate discrepancies in the field. This work will be of broad interest to microbiologists.

**Abstract** Teichoic acids (TA) are linear phospho-saccharidic polymers and important constituents of the cell envelope of Gram-positive bacteria, either bound to the peptidoglycan as wall teichoic acids (WTA) or to the membrane as lipoteichoic acids (LTA). The composition of TA varies greatly but the presence of both WTA and LTA is highly conserved, hinting at an underlying fundamental function that is distinct from their specific roles in diverse organisms. We report the observation of a periplasmic space in *Streptococcus pneumoniae* by cryo-electron microscopy of vitreous sections. The thickness and appearance of this region change upon deletion of genes involved in the attachment of TA, supporting their role in the maintenance of a periplasmic space in Gram-positive bacteria as a possible universal function. Consequences of these mutations were further examined by super-resolved microscopy, following metabolic labeling and fluorophore coupling by click chemistry. This novel labeling method also enabled in-gel analysis of cell fractions. With this approach, we were able to titrate the actual amount of TA per cell and to determine the ratio of WTA to LTA. In addition, we followed the change of TA length during growth phases, and discovered that a mutant devoid of LTA accumulates the membrane-bound polymerized TA precursor.

## Introduction

Teichoic acids (TA) are major components of the cell envelope of Gram-positive bacteria. TA are complex linear polysaccharides attached either to the plasma membrane (lipoteichoic acids, LTA) or to the peptidoglycan (PG) (wall teichoic acids, WTA) (***Brown et al., 2013***; ***Denapaite et al., 2012***; ***Percy and Gründling, 2014***). Either WTA or LTA are dispensable in some laboratory conditions for the growth of *Bacillus subtilis* (***D'Elia et al., 2006a***; ***Schirner et al., 2009***) or *Staphylococcus aureus* (***D'Elia et al., 2006b***; ***Oku et al., 2009***), but at least one type must be present (***Schirner et al., 2009***; ***Oku et al., 2009***). In *Streptococcus pneumoniae*, LTA are dispensable (***Heß et al., 2017***), and reduced

amount of WTA is tolerated (*Minhas et al., 2023*) (strains totally devoid of WTA have not been characterized yet due to partial redundancy of WTA attachment enzymes). TA participate in numerous cellular processes such as cell wall elongation and hydrolysis, cell division, transport processes, cation homeostasis, resistance to antimicrobial peptides, and interactions with the environment or the host.

In studies of Gram-positive bacteria (*B. subtilis, S. aureus, Enterococcus gallinarum*, and *Streptococcus gordonii*) by cryo-electron microscopy of vitreous sections (CEMOVIS; *Dubochet et al., 1988*), a region between the cytoplasmic membrane and the cell wall was proposed to be a periplasmic space (*Matias and Beveridge, 2007*; *Matias and Beveridge, 2008*; *Zuber et al., 2006*). A most interesting speculation is the possible role of TA in maintaining this periplasmic space, so that it would be isotonic with the cytoplasm (*Figure 1A*; *Erickson, 2021*).

In most organisms, LTA and WTA have different compositions and are assembled and exported by different pathways (*Brown et al., 2013*; *Percy and Gründling, 2014*). TA of streptococci of the *mitis* group, such as *S. pneumoniae*, are particular in two ways: LTA and WTA are products of the same biosynthetic pathway and are therefore of very similar composition, and they are uniquely decorated by phosphocholine residues (*Figure 1B*; *Denapaite et al., 2012*).

The role of TA in the pneumococcus is also critical because 13–16 different proteins (choline-binding proteins) are anchored to the cell envelope by the interaction of their choline-binding domain to phosphocholine residues (*Maestro and Sanz, 2016*). The functions of the choline-binding proteins are diverse, several of them are acting on the cell wall itself as hydrolases such as LytA and LytB, others are virulence factors mediating interactions with the host and its immune system.

The pneumococcus has a strict nutritional requirement for choline, which is solely incorporated in its TA (*Tomasz, 1967*). However, numerous derivatives can substitute for choline, as long as the hydroxyl group is present and the amine substituents are not too bulky (*Badger, 1944*). Nutritional shift between choline and ethanolamine had been used in early studies of the localization and composition of nascent cell wall (*Laitinen and Tomasz, 1990*). More recently, we have used the incorporation of alkyne- and azido-choline (aCho) derivatives to label TA with clickable-fluorophores and investigate the spatial and temporal relationship between TA and PG assembly by fluorescence microscopy (*Di Guilmi et al., 2017*; *Bonnet et al., 2018b*).

The pneumococcal repeating unit of TA consists of a pseudo-pentasaccharide ((→4)-6-O-P-Cho-α-D-Gal*p*NAc-(1→3)-6-O-P-Cho-β-D-Gal*p*NAc-(1→1)-Rib-ol-5-P-(O→6)-β-D-Glc*p*-(1→3)-AATGal*p*-(1→)) (*Figure 1—figure supplement 1*; *Fischer, 1997*). The glucose (D-Glc*p*) and the ribitol (Rib-ol) are joined by a phosphodiester bond. The two *N*-acetylgalactosamine (D-Gal*p*NAc) are substituted on position 6 with a phosphocholine, as mentioned above. The phosphocholine is sometimes lacking on the proximal Gal*p*NAc of some subunits, as well as from both Gal*p*NAc from the terminal unit. Subunits are linked together by α-(1→6) bonds. In LTA, the acetamido-4-amino-6-deoxygalactopyranose (AATGal*p*) of the first subunit is β-(1→6)-linked to the glucose of a mono-glucosyl-diacyl-glycerol (DGlc*p*-DAG) (*Gisch et al., 2013*). In WTA, the first AATGal*p* is thought to be α-linked via a phosphodiester to position 6 of the PG *N*-acetyl muramic acid, as in WTA of other species, although this labile linkage has never been experimentally observed in *S. pneumoniae* (*Fischer, 1997*; *Bui et al., 2012*).

Like other surface polysaccharides such as the PG or the capsule, the basic unit is intracellularly synthesized at the plasma membrane onto the carrier lipid undecaprenyl pyrophosphate by a succession of glycosyl- and phosphotransferases (*Denapaite et al., 2012*). The basic unit precursor is then thought to be flipped across the membrane by the flippase TacF (*spr1150*) to allow elongation of the TA polymers at the cell surface by the polymerase TarP (*spr1222*) (*Damjanovic et al., 2007*; *Liu et al., 2017*; *Gibson and Veening, 2023*). Once on the external surface of the membrane, the polymerized precursors can undergo two fates (*Figure 1B*). A phosphotransferase can anchor the polymer onto the PG to yield WTA. Alternatively, a glycosyltransferase can transfer the polymer to DGlc*p*-DAG to form LTA. The phosphotransferase attaching WTA is thought to be LytR (*spr1759*) (*Minhas et al., 2023*), although LytR and the two homologous enzymes Cps2A (*spr 0314*) and Psr (*spr1226*) that form the LCP (LytR-Cps2A-Psr) family may exhibit some redundancy (*Kawai et al., 2011*; *Eberhardt et al., 2012*). The glycosyltransferase producing LTA was found to be TacL (*spr1702*), since the purification procedure normally yielding LTA produced no detectable TA compounds from a Δ*tacL* strain (*Heß et al., 2017*).

Some fundamental aspects of the pneumococcal biology have recently been proposed to rely on shifting the ratio of WTA to LTA. Notably, competence, the process allowing incorporation of DNA

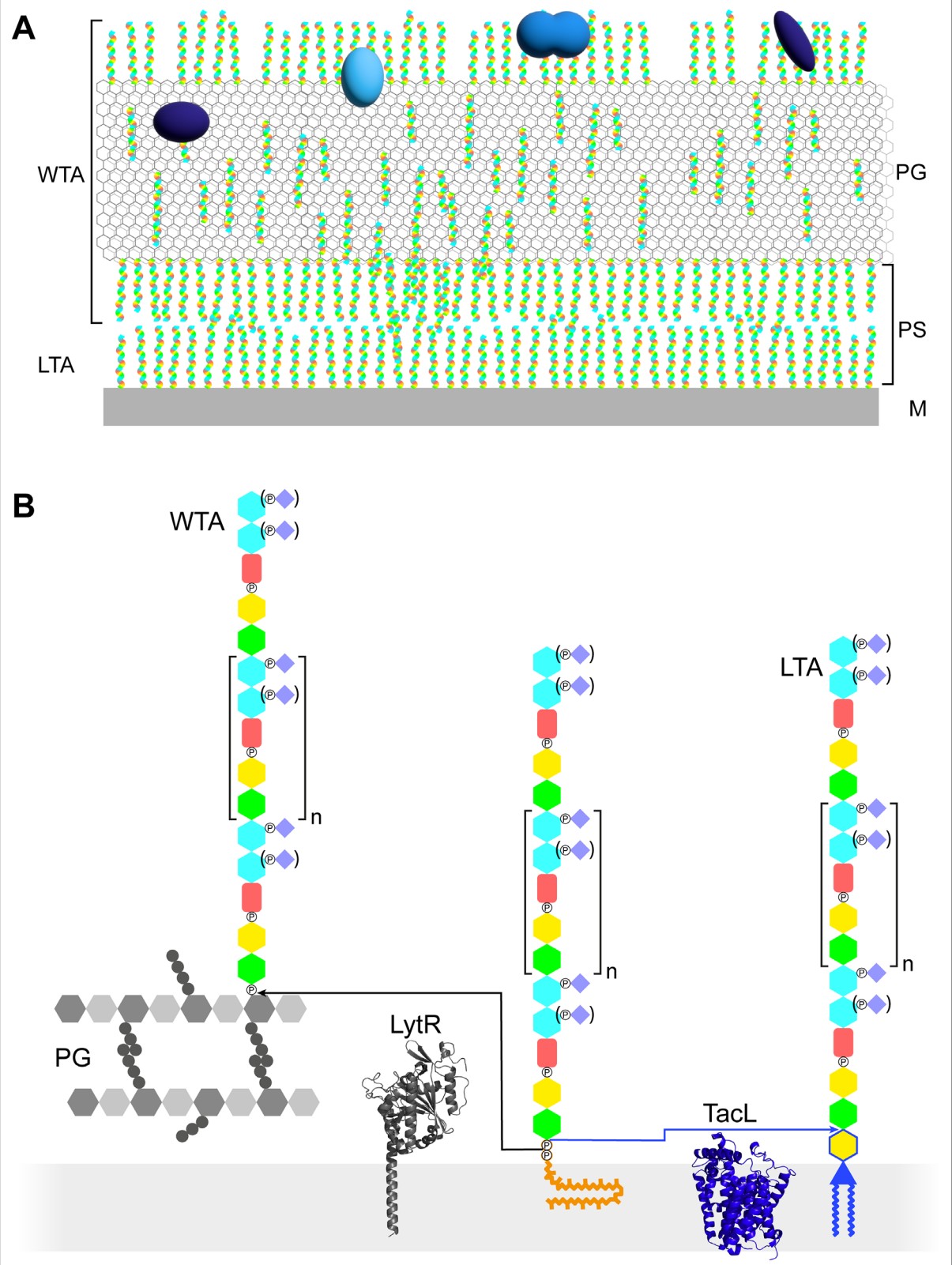

**Figure 1.** Models of the functions and structures of teichoic acids (TA) in *S. pneumoniae*. (**A**) TA (multicolored rods) attached to the membrane (M) or to the peptidoglycan (PG) exclude each other to maintain a periplasmic space (PS) as proposed by *Erickson, 2021*. The sizes of the molecules are not to scale and their density in the different locations is arbitrary. Additional functions arise from the presentation of various choline-binding proteins (blue). (**B**) Polymerized TA attached to undecaprenyl pyrophosphate are transferred at the cell surface onto PG by the phosphotransferase LytR to form wall

*Figure 1 continued on next page*

Figure 1 continued

teichoic acids (WTA); or onto DGlc*p*-DAG by TacL to form lipoteichoic acids (LTA). The enzymes are depicted by their AlphaFold models. The most abundant TA species have 6 repeating units (*n* = 4).

The online version of this article includes the following figure supplement(s) for figure 1:

**Figure supplement 1.** Chemical structure of TA.

from the extracellular environment, was found to be accompanied by an increase of the amount of WTA dependent on the presence of the phosphotransferase LytR, whose expression is upregulated during competence (*Minhas et al., 2023*). In contrast, autolysis, which occurs during the stationary phase of planktonic cultures and is mostly driven by the PG amidase and choline-binding protein LytA, was shown to depend on the relative abundance of LTA and WTA (*Flores-Kim et al., 2019*), modulated by the phosphodiesterase WhyD that releases TA from the PG (*Flores-Kim et al., 2022*).

WTA are attached to the C-6 hydroxyl of *N*-acetyl muramic acids in the PG of *S. pneumoniae*. This position is also the site of *O*-acetylation by Adr (*Crisóstomo et al., 2006*). The same hydroxyl is also often assumed to be the site of attachment of the capsule by a similar phosphodiester linkage (*Eberhardt et al., 2012*), although the serotype 2 capsule has now been found to be attached via a direct glycosidic bond to the C-6 position of PG *N*-acetyl glucosamine (*Larson and Yother, 2017*). There is thus certainly a cross-talk between TA attachment to the PG, *O*-acetylation and capsule attachment. The present study was carried out with the strain R800 as 'wild type' (WT) and derivatives thereof. These strains have a functional *adr* gene but are without capsule.

In this work, using CEMOVIS (cryo-electron microscopy of vitreous thin sections), we report the existence of a periplasmic space in *S. pneumoniae* WT, Δ*tacL*, and Δ*lytR* strains. The observed differences in the appearance and thickness of the periplasmic space supports a participation of TA to its maintenance. The sites of TA incorporation in the cell wall were examined by the metabolic incorporation of aCho and subsequent secondary fluorescent labeling that allowed imaging by super-resolved microscopy, to reveal consequences of the absence of TacL and LytR. In addition, the discovery of the unexpected and previously unreported sedimentation property of the LTA-containing membranes of *S. pneumoniae* allows the simple separation and analysis by gel electrophoresis of labeled LTA and WTA in different strains and conditions. This novel simple method of TA analysis will facilitate studies of TA, notably regarding the ratio of LTA to WTA in various cellular processes.

## Results

### Cryo-electron microscopy analysis of *S. pneumoniae* cell envelope

CEMOVIS is a cellular cryo-electron microscopy method that previously provided evidence for a periplasmic space in several Gram-positive bacteria (*Zuber et al., 2006*; *Matias and Beveridge, 2005*). To investigate the contribution of TA to the architecture of the cell envelope of *S. pneumoniae*, we applied CEMOVIS (*Figure 2—figure supplement 1*) to the WT strain and two mutants thereof thought to be deficient in the attachment of TA to the cell wall (Δ*lytR*) or to the plasma membrane (Δ*tacL*) (*Heß et al., 2017*; *Minhas et al., 2023*).

As in other bacterial species, CEMOVIS reveals the existence of a periplasm in WT *S. pneumoniae*, as evidenced by a light region observed between the plasma membrane and the cell wall (*Figure 2*). Within this light region, a single dark line runs parallel to the membrane. This observation in *S. pneumoniae* supports the idea that a periplasmic space exists in Gram-positive bacteria, and that this space contains organized material, which was previously described as a granular layer (*Zuber et al., 2006*).

If a Gram-positive periplasm filled with TA exists, it must contain LTA, which are anchored in the plasma membrane, and possibly WTA protruding from the inner face of the cell wall layer (*Erickson, 2021*). This hypothesis predicts that variations in LTA and/or WTA incorporation should modify the architecture of the cell envelope. Strikingly, the granular layer present in the WT strain completely disappears in Δ*lytR* and Δ*tacL* cells (*Figure 2*, *Figure 2—figure supplement 2*). In the absence of TacL, the thickness of the periplasmic space is significantly reduced by (34 ± 5)% (SE) compared to that of the WT strain (*Table 1*), supporting the idea that LTA occupy the periplasmic space. Similarly, the absence of LytR induces a significant reduction of the periplasm thickness ((29 ± 5)% (SE), *Table 1*), suggesting that WTA also participate in the maintenance of this region. The thickness of the

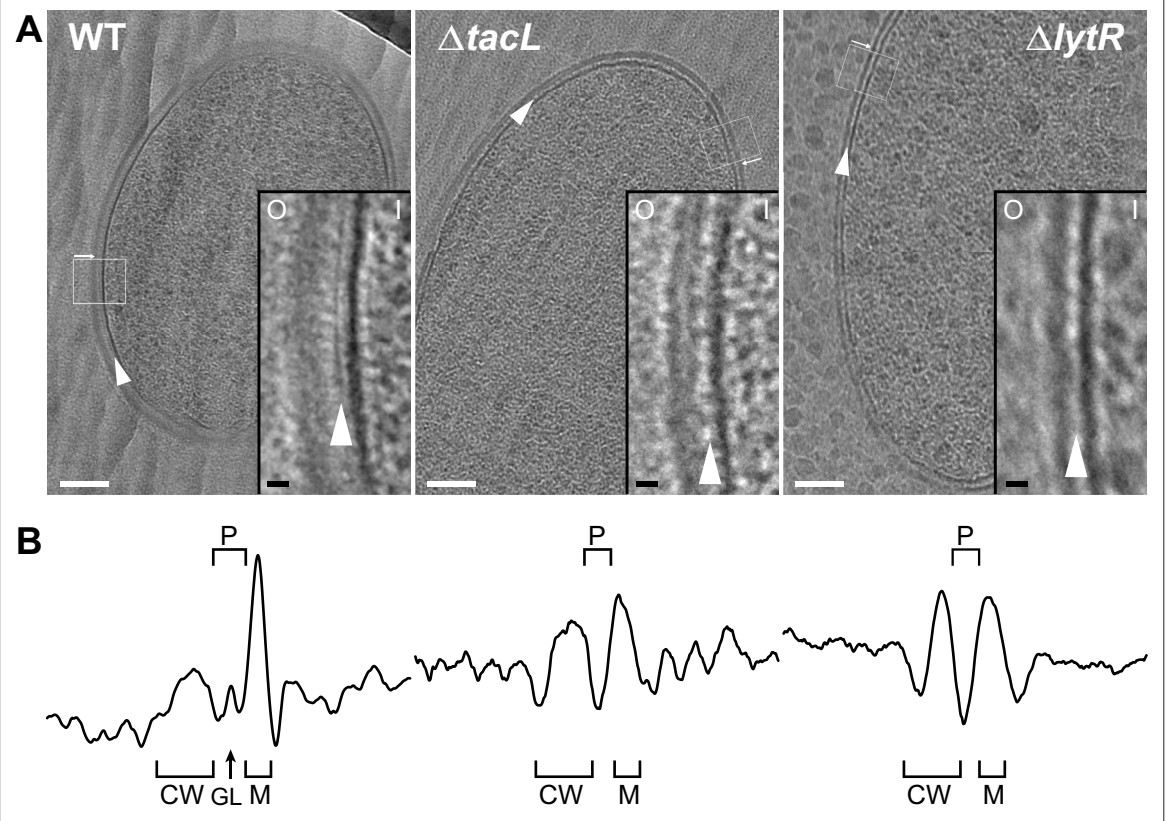

**Figure 2.** Electron micrographs of wild type (WT), Δ*tacL*, and Δ*lytR* strains. (**A**) CEMOVIS micrographs allowing full view (white scale bar, 100 nm) and zoom (black scale bar, 10 nm) of WT and mutant strains. O and I signal the outer and inner sides of the cell envelope. A periplasmic space can be observed in all three strains. A granular layer is seen as a thin black line in the periplasmic space (arrowheads) of WT cells, whereas it is not observed in the Δ*lytR* and Δ*tacL* strains. (**B**) Corresponding pixel intensity profile of the cell envelope is shown for each strain. The intensity profiles were measured perpendicularly to the cell surface in regions boxed in white, and show variations in the density of the envelope ultrastructure. CW, cell wall; P, periplasm; GL, granular layer; M, membrane.

The online version of this article includes the following figure supplement(s) for figure 2:

**Figure supplement 1.** Low-magnification CEMOVIS images of *S. pneumoniae* Δ*tacL* cells.

**Figure supplement 2.** Electron micrographs of wild type (WT), Δ*tacL*, and Δ*lytR* strains.

**Figure supplement 3.** Electron micrographs of pneumococcal cell envelope.

periplasmic spaces in Δ*tacL* and Δ*lytR* strains does not differ significantly from each other. In order to increase the signal-to-noise ratio of our cryo-EM images, we took advantage of the exceptional grid adhesion of one vitreous ribbon of Δ*tacL* cells (*Figure 2—figure supplement 2*) to perform CETOVIS (cryo-electron tomography of vitreous sections). The reconstructed cryo-electron tomogram confirmed the absence of the granular layer throughout the entire volume of a Δ*tacL* cell section (*Figure 2—figure supplement 3A, B*).

Interestingly, CEMOVIS data obtained from the WT, Δ*lytR*, and Δ*tacL* strains also showed that the thickness of the cell wall layer itself is affected when WTA or LTA assembly is impaired (*Figure 2*, *Figure 2—figure supplement 2*, *Table 1*). Indeed, we observed a significant reduction of the cell wall thickness of (17 ± 5)% (SE) or (36 ± 4)% (SE) in the absence of TacL or LytR, respectively. Consistent with the fact that WTA are covalently linked to the PG, the reduction in the cell wall thickness is more pronounced in Δ*lytR* cells compared to Δ*tacL* cells. In contrast to the CEMOVIS method, transmission electron microscopy analysis of sectioned stained freeze-substituted resin-embedded samples (*Figure 2—figure supplement 3C*) did not reveal measurable differences between the cell wall of the WT and mutant strains. The differing appearance of Gram-positive cell envelope using the CEMOVIS and freeze-substitution methods, and the difficulty of evidencing the periplasmic space with the latter, had previously been mentioned and discussed (*Matias and Beveridge, 2005*).

**Table 1.** Dimensions of cell envelope structures measured on CEMOVIS images.
*p=0.05 can be removed from the footnote (twice).

| Strain | Mean thickness (nm) ± SD (nb of measurements)[a] | | | Ratio of thickness CW/PS | GL dist. from membrane[b] (nm) ± SD |
| | CW | PS | GL | | |
|---|---|---|---|---|---|
| WT | 15.9 ± 2.5 (*Trouve et al., 2021b*) | 10.8 ± 1.4 (*Trouve et al., 2021b*) | 3.2 ± 0.8 (*Trouve et al., 2021b*) | 1.5 | 5.0 ± 0.8 (*Trouve et al., 2021b*) |
| Δ*tacL* | 13.5 ± 3.7 (*Trouve et al., 2021b*)[c] **/[d] ** | 6.9 ± 2.3 (*Trouve et al., 2021b*)[c] **/[d] # | NA | 1.95[c] **/[d] ** | NA |
| Δ*lytR* | 10.2 ± 1.7 (*Trouve et al., 2021b*)[c] **/[d] ** | 7.7 ± 2.0 (*Trouve et al., 2021b*)[c] **/[d] # | NA | 1.32[c] #/[d] ** | NA |

[a] Values in nm are presented as mean values ± standard deviations (with numbers of measurements in parentheses). They have been measured using images with a pixel size of 3.7 Å. NA, not applicable.
[b] Measured from the outer surface of the cytoplasmic membrane to the inner most side of the GL.
[c] ANOVA test was performed between corresponding structures of WT and mutant cells. #, no statistical difference; *p = 0.05; **p = 0.01.
[d] ANOVA test was performed between corresponding structures of Δ*lytR* and Δ*tacL* cells. #, no statistical difference; *p = 0.05; **p = 0.01.

## Cellular localization of TA insertion

Intrigued by the effect of the absence of TacL or LytR on the ultrastructure of the cell envelope, we examined how the deletion of *tacL* or *lytR* impacts the localization of TA assembly in *S. pneumoniae* cells. To localize newly synthesized TA, an excess of aCho was added for 5 min to exponentially growing cell cultures. Cells were then fixed prior to secondary labeling by strain-promoted azide-alkyne cyclo-addition (SPAAC) click reaction with the DBCO-linked fluorophore Alexa Fluor 647 (DBCO-AF647), following a protocol adapted from *Trouve et al., 2021a*. As expected, TA insertion occurs at mid-cell (*Figure 3—figure supplement 1*). Although the morphology of Δ*tacL* cells is altered, with pointy poles and some expanded and shrunk cells, enriched aCho incorporation was also detected at midcell. In the Δ*lytR* strain, in contrast to the WT parental and Δ*tacL* strains, the 5-min incubation with aCho yielded only a weak mid-cell labeling in few cells (*Figure 3—figure supplement 1*). The weak labeling of Δ*lytR* cells may reflect one or a combination of the following: (1) TA are less abundant, (2) cells incorporate less aCho into TA, or (3) cells grow more slowly, so that the 5-min pulse represents a shorter fraction of their cell cycle.

The photophysical properties of AF647 are amenable to direct stochastic optical reconstruction microscopy (dSTORM), which allowed the reconstruction of super-resolved images as presented in *Figure 3*, showing the localization of fluorescent TA after a 5-min labeling pulse experiment. Corresponding images in conventional fluorescence and bright-field mode are shown in *Figure 3—figure supplement 2*. In the three strains, TA were mostly inserted at mid-cell, where the PG is assembled (*Trouve et al., 2021b*). However, some localizations were also detected elsewhere at the cell surface in all three strains. This heterogenous labeling was similar in all three strains, rendering the weak labeling at the division site of Δ*lytR* cells difficult to identify. This is in contrast to what is observed with PG insertion, where very little signal is observed outside of the cell division zone (*Figure 3* and *Trouve et al., 2021b*). The localization patterns of newly incorporated TA observed in these experiments are mostly that of WTA, since this distribution is preserved in sacculi preparations (*Figure 3—figure supplement 2B*). The bright spots that were often seen in sacculi likely correspond to aggregated material that is trapped inside unbroken sacculi.

While there is no discernible difference between the WT and Δ*tacL* cells in the distribution of TA labeled during a 5-min pulse, the evolution of this distribution following a chase, when growth is pursued for up to 35 min (about one WT generation time) without aCho, is markedly different between the two strains (*Figure 3*). In the WT strain, the separated banded pattern expected from the insertion of unlabeled TA at the division site is observed, mirroring the patterns produced by PG labeling. After 35 min of chase, most WT cells show polar or equatorial labelings following completion of their division. In contrast, in Δ*tacL* cells, labeled TA can be seen over the whole new cell surface that is generated during the pulse and the first 15 min of chase. An unlabeled region is visible at mid-cell in Δ*tacL* cells only after 35 min of chase. The chased pattern of labeled PG in the Δ*tacL* strain is not affected and is similar to that in WT.

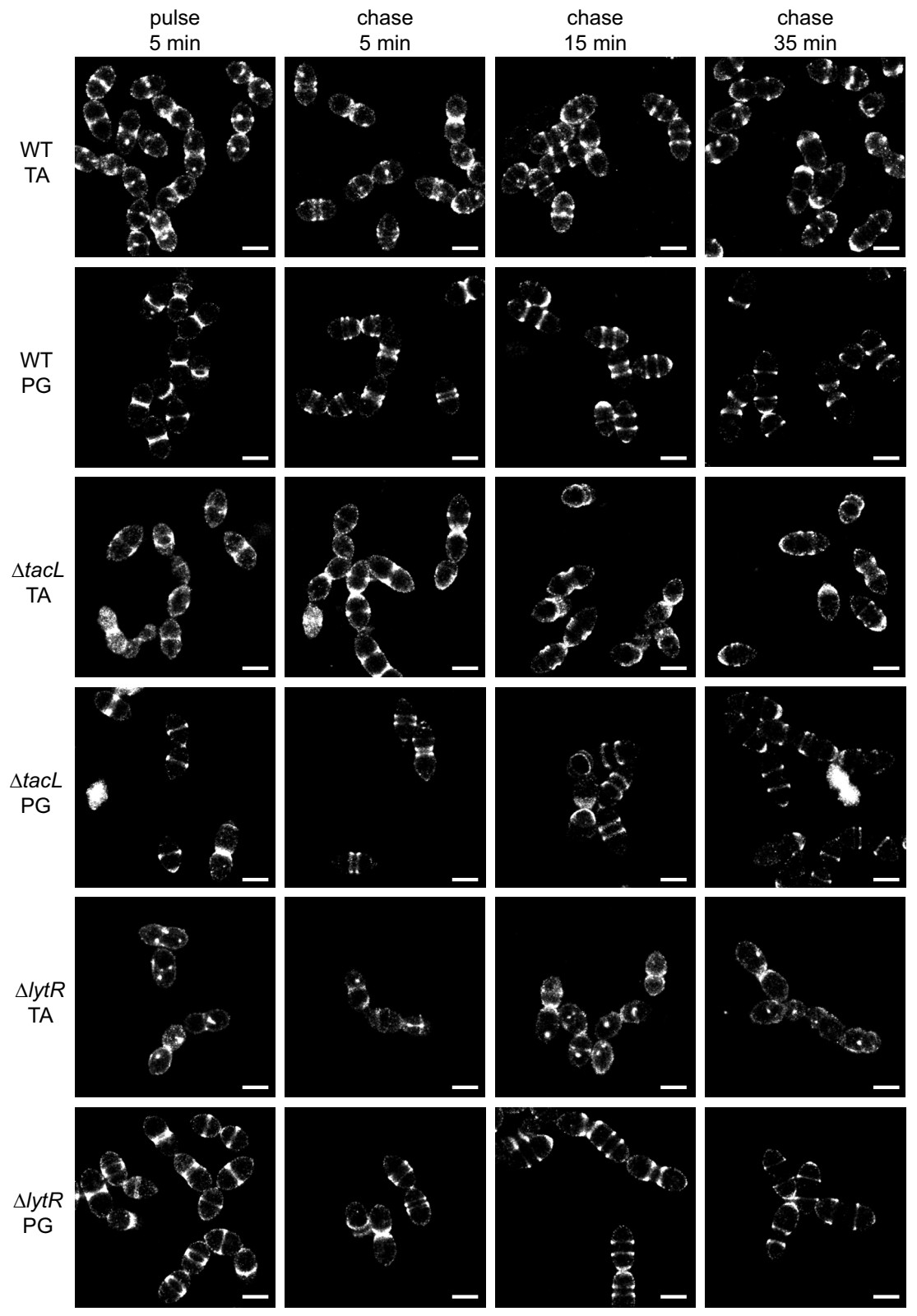

**Figure 3.** Super-resolved direct stochastic optical reconstruction microscopy (dSTORM) imaging of teichoic acids (TA) incorporated in growing wild type (WT), Δ*tacL*, and Δ*lytR* cells after a 5-min pulse of metabolic labeling with 1.5 mM aCho followed by a chase, and subsequent secondary fluorescent labeling by click chemistry using DBCO-AF647. For comparison, the same pulse–chase procedure was applied to reveal the newly synthesized

*Figure 3 continued on next page*

*Figure 3 continued*

peptidoglycan (PG) with a metabolic labeling using 2 mM azido-D-Ala-D-Ala. Scale bars are 1 μm. Corresponding conventional and bright-field images are shown in *Figure 3—figure supplement 2*.

The online version of this article includes the following figure supplement(s) for figure 3:

**Figure supplement 1.** Fluorescence microscopy of wild type (WT), ΔtacL, and ΔlytR cells grown in BHI and pulse-labeled with 1.5 mM aCho for 5 min prior to fixation and secondary click-labeling with DBCO-AF647.

**Figure supplement 2.** Fluorescence microscopy of TA labeling.

In dSTORM imaging of the Δ*lytR* strain, the patterns are difficult to discern and many cells show no specific mid-cell labeling. When visible, the labeled area extends during the chase, as described above for the Δ*tacL* strain.

## Separation of labeled LTA and WTA by low-speed centrifugation

In previous studies, unlabeled TA from the pneumococcus have been analyzed by polyacrylamide gel electrophoresis in the presence of sodium dodecyl sulfate (SDS) (SDS–PAGE). Purified WTA were revealed in gel by an Alcian blue staining procedure, whereas LTA from crude membrane fractions were revealed by immune-blotting with anti-phosphocholine antibodies (e.g. *Flores-Kim et al., 2019*; *Flores-Kim et al., 2022*). The standard preparative purification of TA from *S. pneumoniae* starts with the sedimentation of mechanically broken cell walls, which contain the WTA, with the concomitant detergent solubilization of LTA (e.g. *Heß et al., 2017*). Alternatively, for analysis of LTA by immuno-blotting for example, protoplasts prepared by PG enzymatic digestion would be osmotically lysed, cellular debris would be sedimented by low-speed centrifugation and discarded, and LTA-containing membranes would finally be collected by ultracentrifugation (e.g. *Flores-Kim et al., 2019*; *Flores-Kim et al., 2022*). On the other hand, WTA can be released from cell wall fragments by thorough digestion of the PG (e.g. *Heß et al., 2017*; *Bui et al., 2012*) or alkaline hydrolysis (e.g. *Flores-Kim et al., 2019*; *Flores-Kim et al., 2022*).

Since we were able to fluorescently label TA, we thought it would allow simpler in-gel analysis. WT cells were cultured over two generations in C-medium devoid of choline but supplemented with 200 μM aCho. Cells having thus metabolically incorporated aCho were lyzed by overnight digestion of the cell wall with lysozyme, mutanolysin and recombinant LytA. The aCho was coupled to a DBCO-fluorescent Alexa dye (AF488) by SPAAC click reaction during the lysis incubation. The lysates were then cleared of debris by a low-speed centrifugation at $10,000 \times g$, and ultracentrifugation of the supernatant at $100,000 \times g$ was subsequently performed to separate the membrane from the soluble fraction. Since at this stage the TA were fluorescently labeled, we were surprised to find more fluorescence in the low-speed pellet than in the pellet resulting from the ultracentrifugation. Analysis of the soluble fraction (the supernatant) by SDS–PAGE showed the pattern expected from labeled WTA migrating as a set of poorly resolved bands with low electrophoretic mobility, as in typical Alcian blue-stained gels (*Flores-Kim et al., 2019*; *Flores-Kim et al., 2022*; *Figure 4A*). The pattern expected from labeled LTA, a ladder of well-resolved bands with a greater electrophoretic mobility than the WTA, was observed chiefly in the low-speed pellet. Centrifugation of cells lysates with labeled TA at different speeds confirmed that 2 min at $20,000 \times g$ were sufficient to sediment most of the LTA (*Figure 4A*). The nature of high-mobility species and other details on the fractionation and the electrophoresis are commented in the SI (Notes on the fraction of LTA and WTA, *Figure 4—figure supplement 1*; Notes on the electrophoresis, *Figure 4—figure supplement 2*).

Low-speed centrifugation pellets were not transparent and had a loose consistency, different from unbroken cell pellets or classical compact membrane pellets sedimented at $100,000 \times g$. Coomassie blue staining of electrophoresis gels of the supernatant and pellets of low-speed centrifugation showed that the pellet and supernatant have different protein profiles (*Figure 4—figure supplement 3A*). After chloroform extraction, iodine-stained thin layer chromatography showed that the low-speed pellet contained most of the lipids (*Figure 4—figure supplement 4A*) and negative stain transmission electron micrograph of the low-speed pellet revealed membranous material (*Figure 4—figure supplement 4B*). Taken together, these observations indicate that the LTA-containing membrane fraction of lysed *S. pneumoniae* cells sediments when submitted to low relative centrifugal force.

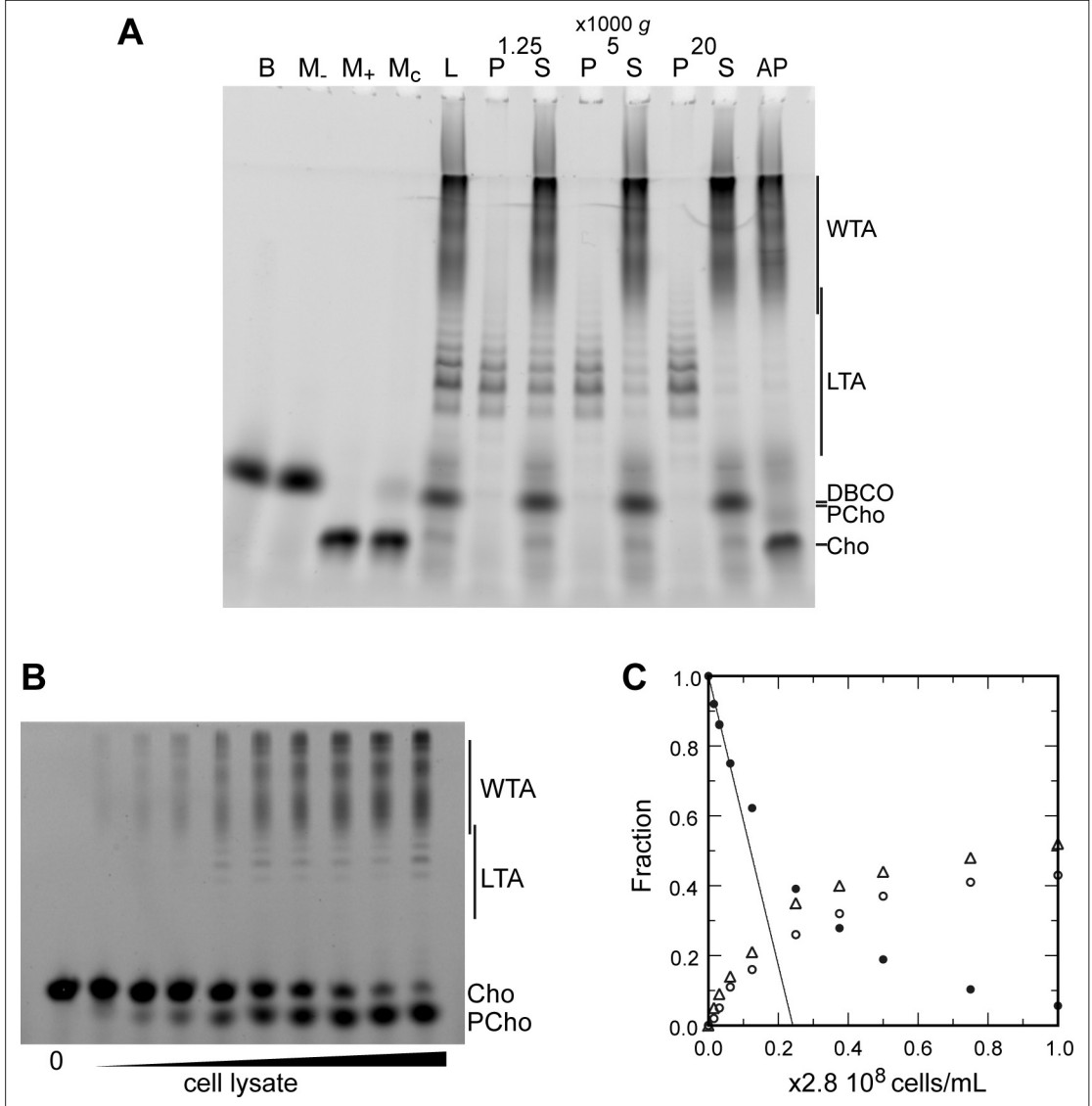

**Figure 4.** Electrophoretic analysis of TA. (**A**) Polyacrylamide gel electrophoresis of fluorescently labeled teichoic acids (TA) demonstrating separation of lipoteichoic acids (LTA) and wall teichoic acids (WTA) by centrifugation. Labeled compounds were revealed by UV transillumination. TA were labeled by growing wild type (WT) cells in C-medium in the presence of 200 µM aCho. Cells were lysed overnight with lysozyme, mutanolysin and LytA and the azido-groups were modified by reaction with 25 µM DBCO-AF488. The lysate was centrifuged for 2 min at 1250, 5000, or 20,000 × *g*. The pellets were resuspended in the initial volume. An aliquot of the 20,000 × *g* supernatant was treated with alkaline phosphatase. The buffer with the lysing enzymes, the medium with and without aCho and the culture supernatant were similarly incubated with 25 µM DBCO-AF488. B, lysis buffer with enzymes; M−, C-medium without aCho; M+, C-medium with aCho; M$_C$, culture supernatant; L, lysate; P, pellet; S, supernatant; AP, alkaline phosphatase-treated. The control samples (B, M−, M+, and M$_C$ were loaded at one-fourth the volume of the cellular samples). Labels on the right side of the gel identify species coupled to the fluorophore AF488. (**B, C**) Titration of cellular TA. WT cells were grown for two generation in the presence of aCho prior to cell lysis. DBCO-AF488 (1.9 µM) was incubated for 24 hr with varying amounts of cell lysate corresponding to up to $2.9 \times 10^8$ cells·ml$^{-1}$. The remaining DBCO-AF488 was blocked by addition of 100 mM aCho, and the various species were separated by polyacrylamide (17%) gel electrophoresis (**B**). The bands were quantified and the relative amount of the various species were plotted against the cell concentration (**C**). Black circles, blocked DBCO-AF488; open circle, phosphocholine; open triangle, TA. A linear regression of the DBCO-AF488 points at low cellular concentration was applied to obtain the titration point as the intercept of the cell concentration axis.

The online version of this article includes the following figure supplement(s) for figure 4:

**Figure supplement 1.** Electrophoretic analysis of TA and membrane preparations.

**Figure supplement 2.** Polyacrylamide gel electrophoresis of click-labeled teichoic acids (TA), aCho, and the clickable fluorophore DBCO-AF488 with (**A**) 12% (wt/vol) and (**B**) 17% acrylamide.

**Figure supplement 3.** Coomassie staining of TA analysis gels.

*Figure 4 continued on next page*

## NMR analysis of LTA prepared after low-speed centrifugation

To determine that the low-speed sedimentation of LTA-containing membranes was not an artifact of the labeling method, we applied the same separation method to unlabeled cells followed by the extraction procedure previously reported (*Heß et al., 2017*) for NMR analysis. The $^{31}$P NMR spectrum showed the presence of the typical TA phosphorus peaks from internal and terminal units (*Figure 4—figure supplement 5*). Subsequent treatment with hydrazine to remove acyl chains improved the spectra as expected (*Gisch et al., 2013*). Natural abundance $^{13}$C, $^{1}$H-HSQC spectrum showed the presence of the expected sugar species, including the glucosyl-glycerol from the lipid anchor of LTA (A and GRO peaks, *Figure 4—figure supplement 5*).

## Titration of cellular TA

The fluorescent labeling of TA by click chemistry and separation of LTA and WTA allowed the titration of the different species in cells. For this, we incubated varying quantities of the aCho-labeled cell lysate with fixed amounts of DBCO-AF488. Samples were analyzed by gel electrophoresis and the relative amount of the various fluorescent species was determined by densitometry and plotted against the concentration of cells. The gels and quantifications are shown in *Figure 4B, C*, *Figure 4—figure supplement 6A, B* for two concentrations of DBCO-AF488. For intermediate amounts of lysate, the concentrations of clickable groups were too low to reach reaction completion during the incubation time. Only with the highest amount of lysate could the fluorophore be nearly consumed. However, for low amounts of lysate, when the fluorophore is in large excess, the reaction is also expected to have neared completion. Therefore, the fraction of unreacted DBCO-AF488 was linearly extrapolated to zero from the measurements at low lysate concentrations, to obtain the amount of lysate that would titrate the fluorophore in an infinite reaction time. Thus, 1.9 μM DBCO-AF488 would titrate the metabolized aCho from $(0.24 \pm 0.01) \times 2.8 \times 10^8$ cells·ml$^{-1}$ and 3.7 μM would titrate aCho from $(0.45 \pm 0.01) \times 2.8 \times 10^8$ cells·ml$^{-1}$ (*Figure 4B, C*, *Figure 4—figure supplement 6A, B*), which is equivalent to $(17 \pm 2) \times 10^6$ incorporated aCho molecules per cell. Since about 55% of labeled choline are incorporated into TA, the rest being found in the form of phosphocholine, there are about $(9.4 \pm 1.1) \times 10^6$ labeled TA choline per cell. Over two generations grown in the exclusive presence of labeled choline, at most 75% of TA choline residues are expected to be labeled, as measured previously (*Di Guilmi et al., 2017*). The total amount of TA choline residues per cell can therefore be estimated to be about $(12.5 \pm 1.5) \times 10^6$. Assuming that the major species of TA consists of 6 repeating units, as determined previously (*Gisch et al., 2013*; *Bui et al., 2012*), with each unit comprising about two choline residues, the number of TA molecules per cell is about $10^6$.

Although the electrophoretic mobility of the longest species of LTA overlaps that of the shortest WTA species, the major species are well separated (*Figure 4B, C*, *Figure 4—figure supplement 6A, B*), allowing their relative quantification. Labeled choline residues incorporated into LTAs were found to account for $15 \pm 5\%$ of the TA choline residues. Assuming that the amount of polymerized TA precursors is negligible, and that size distribution is similar for LTA and WTA, LTA and WTA represent roughly 15 and 85% of pneumococcal TA, respectively.

## Modification of TA length in different *S. pneumoniae* growth phases

Using TA labeling and sedimentation separation, we examined whether changes in the distribution of TA could be detected during growth. WT and Δ*lytA* cells were grown with aCho over two generations. LytA is the main autolysin of *S. pneumoniae*, and culture of cells lacking this PG hydrolase show an extended stationary phase with minimal lysis. Cells were harvested during the exponential phase of growth, at the onset of the stationary phase, and when the WT cells started to lyse. Remarkably, longer species of LTA can be observed at later stages of growth (*Figure 5A, B*, *Figure 5—figure supplement 1A*). Assuming that the most abundant form consists of 6 repeating units (*Gisch et al., 2013*), the 5- and 7-unit species are the next most abundant species, and traces of the 8- and 9-unit

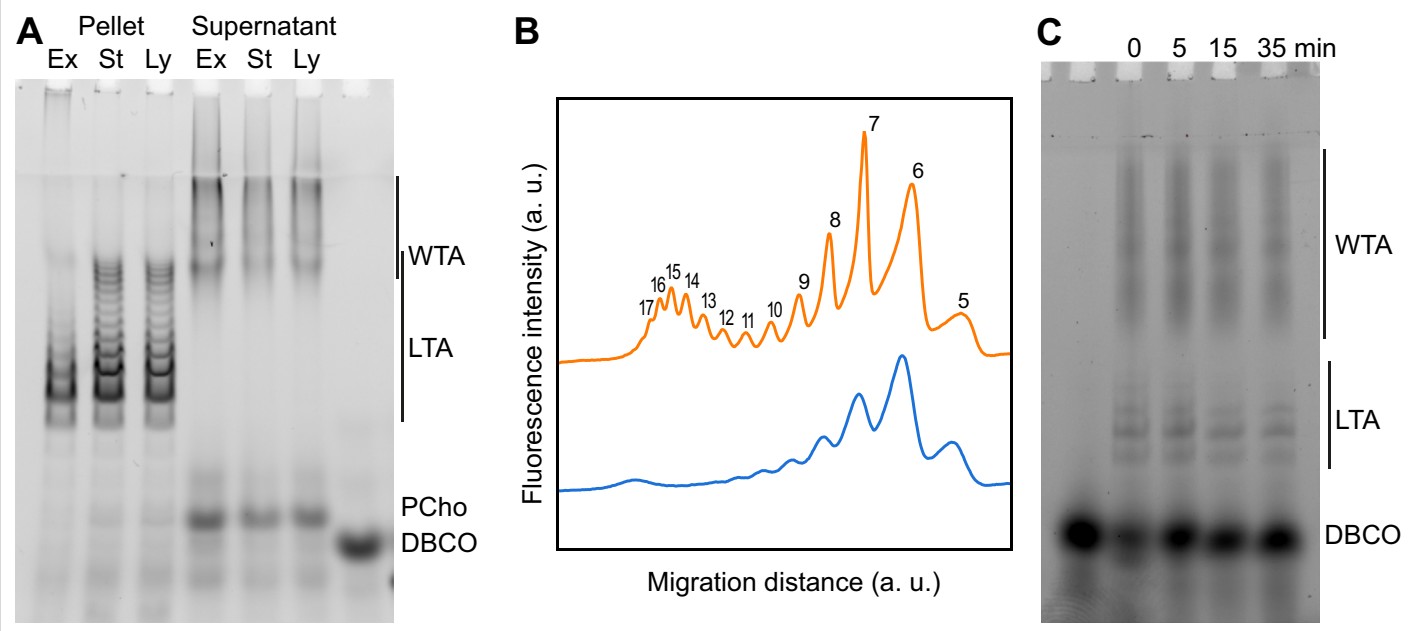

**Figure 5.** Modification of teichoic acids (TA) length in different growth phases. (**A**) Wild type (WT) cells grown in C-medium containing 200 µM aCho were harvested during the exponential growth phase (Ex), at the onset of the stationary phase (St), and during the autolysis (Ly). TA were fluorescently click-labeled with DBCO-AF488 and cells were completely lysed by the addition of peptidoglycan (PG) hydrolases prior to centrifugation. Lipoteichoic acids (LTA) are found in the pellet (P) whereas wall teichoic acids (WTA) are observed in the supernatant (S). The amounts of cells at the different culture stages were normalized. The LTA samples are eightfold concentrated compared to the WTA samples. Fluorescently labeled TA were revealed by UV transillumination after sodium dodecyl sulfate (SDS)–polyacrylamide electrophoresis. (**B**) Densitometric profiles of the fluorescent intensities of exponential (blue) and stationary phase samples (orange) shown in (**A**). (**C**) Electrophoretic analysis of TA of WT cells grown in C-medium with 0.1% yeast extract and pulse-labeled for 5 min with 200 µM aCho, and chased by further growth in the same medium without added aCho for the indicated duration.

The online version of this article includes the following figure supplement(s) for figure 5:

**Figure supplement 1.** Modification of teichoic acids (TA) during growth.

species can be detected during the exponential phase of growth. Later during the culture, although the 6-unit species remains the most abundant and the 5-unit is detected at level comparable as earlier in the culture, longer species become more abundant with up to 17 identifiable units. Surprisingly, the size distribution appears to be bimodal with a first group of species consisting of 5–11 units, and a second group of longer forms where the most abundant one contains 15 units. When considering the electrophoretic pattern, it should be considered that the signal of the different species is proportional not only to their amount but also to their length, since longer species harbor more labeled choline residues. Therefore, long species are less abundant than they appear.

There was no significant difference between the strains with and without LytA (*Figure 5A, B*, *Figure 5—figure supplement 1A*). The ΔlytA did not lyse spontaneously, but a similar amount of TA with the same distribution was obtained from the lysing lytA$^+$ culture. Thus, although the WT cells lyse, their WTA and LTA are not degraded and remain unchanged in the medium.

We also monitored the fate of the labeled species during a chase period of further growth following a 5-min labeling pulse of WT cells, mirroring the dSTORM observations. No modification of the pulse-labeled TA could be detected during the chase time, whether the growth, labeling pulse and chase were performed in BHI or C-medium and supplemented with yeast extract (*Figure 5C*, *Figure 5—figure supplement 1B*).

## LTA and the role of TacL

WTA and LTA were examined in a strain devoid of TacL, which is thought to be responsible for the attachment of LTA to membrane glycolipids (*Figure 6A*). Labeled TA were found in the pellet, and when analyzed by gel electrophoresis were indistinguishable from LTA. After double checking that

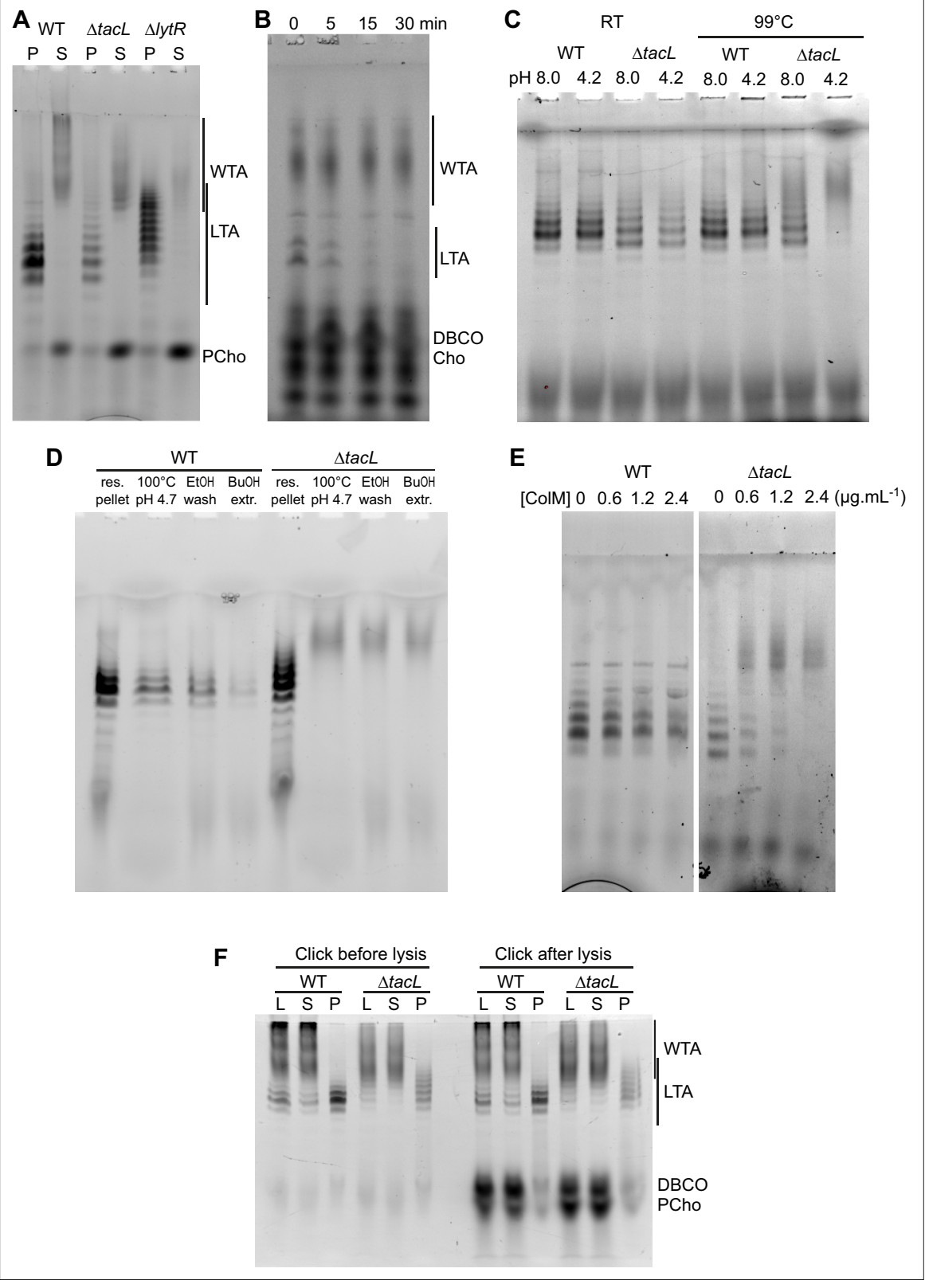

**Figure 6.** Electrophoretic analysis of labeled TA. (**A**) Electrophoretic analysis of teichoic acids (TA) from the wild type (WT) strain, or deleted of the gene encoding the glycosyltransferase TacL, or the phosphotransferase LytR. Cells were grown in C-medium with no choline and 200 μM aCho, prior to cell lysis with peptidoglycan (PG) hydrolases and secondary labeling with DBCO-AF488. Lysates were fractionated by centrifugation and the pellets were resuspended in one fourth the initial volume: P, pellet; S, supernatant. (**B–E**) Identification of sedimented TA from lysates of Δ*tacL* cells as undecaprenyl

*Figure 6 continued on next page*

*Figure 6 continued*

pyrophosphate bound precursors. (**B**) Pulse–chase labeling experiment of TA in strain Δ*tacL*. Cells grown in BHI were exposed to 1.5 mM aCho for 10 min (pulse), prior to washing and further incubation in the same media without aCho (chase) for various time prior to cell lysis and secondary labeling with DBCO-AF488. Samples were treated with alkaline phosphatase prior to electrophoresis. (**C**) Pellets containing labeled TA from WT and Δ*tacL* cells were resuspended in a Tris- or ammonium acetate-buffered solution at pH 8 or pH 4.2 and incubated for 3 hr at room temperature or 99°C prior to electrophoresis. (**D**) Labeled TA from WT and Δ*tacL* cells were submitted to the first steps of the traditional procedure of lipoteichoic acids (LTA) isolation: boiling in 4% sodium dodecyl sulfate (SDS) in a citrate-buffered solution at pH 4.7, lyophilization and ethanol wash of the lyophilizate, butanol extraction and retention of the aqueous fraction. (**E**) Labeled sedimented TA from WT and Δ*tacL* cells were incubated overnight without or with various concentrations of recombinant *P. aeruginosa* colicin M prior to electrophoretic separation. (**F**) To probe on which side of the plasma membrane the TA are positioned, secondary click-labeling with DBCO-AF488 was performed prior or after cell lysis of WT and Δ*tacL* cells grown in C-medium in the presence of 0.2 mM aCho. L, whole lysate; S, supernatant of 20,000 × *g*, 2-min centrifugation; P, pellet resuspended in one-fourth initial volume.

The online version of this article includes the following figure supplement(s) for figure 6:

**Figure supplement 1.** Quantification of the relative amount of wall teichoic acids (WTA) and lipoteichoic acids (LTA) in the Δ*lytR* strain.

**Figure supplement 2.** Growth of the *S. pneumoniae* wild type (WT) (black), Δ*lytA* (gray), Δ*tacL* (red), and Δ*lytR* (blue) strains in BHI (**A**) and C-medium containing 200 µM choline (**B**).

the strain was indeed deleted for the *tacL* gene, we hypothesized that the sedimented TA could be membrane-bound precursors still attached to the undecaprenyl pyrophosphate lipid carrier. In support of this idea, pulse-labeled LTA-like bands disappeared during a chase (*Figure 6B*), as would be expected if these compounds were precursors being transformed into WTA. This is in contrast to the WT strain (*Figure 5C*), in which labeled mature LTA are detected throughout a chase period.

The pyrophosphate linkage in the PG precursor lipid II is labile at low pH and high temperature (*Higashi et al., 1967*). Reasoning that the same should be true for the TA precursor, we incubated the sedimented fraction from the WT and Δ*tacL* strain at 99°C at either pH 8 or pH 4.2. The TA sedimented from the Δ*tacL* lysate were degraded at both pH, whereas those from the parental strain were still detected (*Figure 6C*). This observation suggested that the previously reported failure to isolate TA from the membrane fraction of a Δ*tacL* strain could be due to the process that involves boiling at pH 4.7 (*Heß et al., 2017*). To test this hypothesis, we submitted resuspended pellets of labeled TA from the WT and Δ*tacL* strains to the initial steps of the reported isolation procedure for LTA: boiling in SDS at pH 4.7, lyophilization and ethanol wash of the lyophilizate followed by butanol extraction and retention of the aqueous fraction (*Heß et al., 2017*). Although material was lost at each step, LTA from WT cells survived the procedure unscathed, whereas pelleted TA from the Δ*tacL* strain was lost at the first step, which involves boiling at low pH (*Figure 6D*). In that case, labeled material with a lower electrophoretic mobility appeared as a smear like that of solubilized WTA.

To confirm that the TA sedimented from a lysate of Δ*tacL* cells are membrane-bound via a pyrophosphate, we incubated these TA with recombinant colicin M from *Pseudomonas aeruginosa*. Colicin M are toxins that kill bacteria by hydrolyzing the ester bond between the undecaprenol and the pyrophosphate of the PG precursors lipid I and II (*Chérier et al., 2021*). Colicin M from *P. aeruginosa* was shown to be the most active in vitro and to display some activity on undecaprenyl pyrophosphate without saccharide, thus being less stringent regarding its substrates than other colicin M (*Barreteau et al., 2009*). *P. aeruginosa* colicin M could indeed hydrolyze labeled sedimented TA from Δ*tacL* cells, whereas it was inactive on labeled LTA from WT cells (*Figure 6E*). The colicin M hydrolytic activity was inhibited by chelation of divalent cation by EDTA, as expected (*Barreteau et al., 2009*).

The combined results show that TacL is indeed the glycosyltransferase responsible for the attachment of TA to the membrane glycolipids (*Heß et al., 2017*). Moreover, they reveal that in the absence of TacL, membrane-bound TA remain present in abundance in the form of undecaprenyl-pyrophosphate-attached precursors. These precursors were observed in an early study of the deletion of *tacL*, then known as *rafX*, although there was some confusion at the time between LTA and WTA (*Wu et al., 2014*).

To evaluate if the membrane-bound TA precursors accumulating in the Δ*tacL* strain are located on the inner or outer side of the plasma membrane, cells were grown in C-medium in the presence of 0.2 µM aCho, and the secondary labeling with DBCO-AF488 was performed by a 2-hr incubation at room temperature either before, or after cell lysis. Samples were analyzed by gel electrophoresis (*Figure 6F*). LTA in the WT strain or membrane-bound precursors in the Δ*tacL* strain were equally labeled before and after lysis, like WTA. In contrast, cytoplasmic azido-phosphocholine was

only labeled after lysis, demonstrating that the DBCO-AF488 dye is membrane impermeant. These observations are in agreement with the polymerization of TA taking place at the cell surface (*Gibson and Veening, 2023*).

## LytR is the main WTA phosphotransferase

Previous work indicated that LytR is the major enzyme mediating the final step in WTA formation (*Minhas et al., 2023*). As shown in *Figure 6A*, the proportion of WTA decreased in the ΔlytR strain. However, there was still WTA present, indicating that perhaps another LCP protein can also produce WTA. To further evaluate the contribution of LytR in WTA assembly, the relative amount of WTA and LTA in WT and ΔlytR cells was determined by densitometry of in-gel fluorescence, following growth in C-medium supplemented with 200 µM aCho, or pulse labeling in BHI with 1.5 mM aCho (*Figure 6— figure supplement 1*). In the WT strain, labeled WTA in this particular experiment was found to represent ~80% of the total labeled TA, whereas in the ΔlytR strain, this proportion decreased to ~40%. Thus, the ratio of WTA to LTA amounts is decreased about five to sevenfold in the absence of LytR.

In C-medium, the ΔlytR strain grew more slowly and reached a lower cell density before lysis than its parental strain (*Figure 6—figure supplement 2*). While the WT cells were harvested during their exponential phase of growth, this could not be achieved with ΔlytR cells that were collected at the end of their growth phase to obtain sufficient material for analysis. The longer LTA observed in the ΔlytR strain may therefore result from the cells entering the stationary phase (like the WT strain in *Figure 5A*), rather than from the absence of LytR.

## Discussion

Multiple roles have been found for TA in various bacteria. These varied functions, however, fail to explain why the presence of LTA and WTA with different compositions is conserved in Gram-positive organisms. An underlying common function based on the attributes shared by TA in different species must lie at the root of the conservation of these cell wall polymers through evolution. The hypothesis proposed by Erickson posits that TA are necessary to maintain a periplasmic space at an osmotic pressure that counterbalances that of the cytoplasm (*Erickson, 2021*). TA being maintained for this reason may then have been coopted by evolutionary process to carry out additional functions, like evolutionary spandrels (*Gould and Lewontin, 1979*). This attractive hypothesis is however difficult to test, not least because data supporting the existence of a periplasmic space in Gram-positive bacteria are scarce.

Here, our CEMOVIS images of *S. pneumoniae* (*Figure 2*, *Figure 2—figure supplement 2* and *Figure 2—figure supplement 3A, B*) reveal the presence of a region of lower electron density between the membrane and the PG, as reported previously in other Gram-positive organisms and defined as periplasmic space (*Matias and Beveridge, 2007*; *Matias and Beveridge, 2008*; *Zuber et al., 2006*). About halfway between the PG and the membrane, this periplasmic space appears to be partitioned by a thin layer of more electron-dense material, termed the granular layer in a previous study of *S. gordonii* (*Zuber et al., 2006*). By contrast, freeze-substitution and osmium staining produced thin section of cells with no visible periplasm (*Figure 2—figure supplement 1*). However, an alternative staining procedure after freeze-substitution and thin sectioning, reported elsewhere for encapsulated *S. pneumoniae* D39, yielded images of cells with a three-layered cell envelope (not counting the capsule). It is likely that the middle envelope layer observed in these images is the periplasmic space (*Liu et al., 2017*; *Gallay et al., 2021*). Since high-pressure frozen cells are directly sectioned for CEMOVIS but undergo numerous subsequent treatments for freeze-substitution and contrasting, it is tempting to consider that the CEMOVIS images yield more truthful representations of the cellular architecture. The thinning of the periplasmic space and disappearance of the granular layer in strains with diminished amounts of either WTA (ΔlytR) or LTA (ΔtacL) support Erickson's hypothesis that a periplasmic space is maintained by the physical properties of WTA emanating from the PG layer and LTA springing from the membrane *Erickson, 2021*. In this framework, it is also striking that the appearance of the cell envelope of *S. gordonii* (*Zuber et al., 2006*) and *S. pneumoniae* are so similar, despite the fact that the composition of their TA are very different (*Denapaite et al., 2012*; *Park et al., 2020*; *Lima et al., 2019*), lending credence to a fundamental function of TA stemming from their general physico-chemical properties, rather than from specific molecular interactions.

Erickson listed four physical mechanisms of the polyelectrolyte brush model of TA that could explain the formation of a periplasmic space where TA chains closely packed on a surface of the plasma membrane generate a pressure when pushed against another TA covered surface (the PG layer). These four contributing factors are: '(i) excluded volume, where the chains cannot occupy the same space; (ii) electrostatic repulsion within and between chains; (iii) reduced entropy of the chains as they are confined to a narrow cylinder; (iv) reduced entropy of the counterions as they are concentrated near the anionic charges' (*Erickson, 2021*). Although generally applicable, the electrostatic repulsion experienced by most anionic TA cannot apply to TA from *S. pneumoniae*, since these are zwitterionic and polyampholytes. Indeed, the phosphocholine side chains carry both a positive and a negative charge at physiologically relevant pH (zwitterionic), while the main chain repeated unit also carry one negative chain from its phosphodiester linkage and one positive charge from the amino-group of its AATGal*p* ring (polyampholyte character). Although globally neutral, the local charges of pneumococcal TA may attract counterions, which could also contribute to the osmolality and pressure of the periplasmic compartment. Together with the excluded volume and reduced entropy of TA due to confinement of the neutral brush polymer theory (*de Gennes, 1987*), these would be the main drivers of the maintenance of a periplasmic space in streptococci of the *mitis* group. Since *S. pneumoniae* and *S. gordonii* display similar periplasmic spaces when observed by CEMOVIS, despite having TA with different electrostatic properties, future comparisons of these species in different pH, osmolality, and ionic strength regimen may provide clues about the physico-chemical interactions of these polymers.

The brush polymer model of TA function in forming a periplasmic space applies only if they are present at a density at which volume exclusion occurs. Our metabolic labeling method using aCho that can be titrated with a clickable fluorescent probe has allowed to evaluate the amount of TA chains per cell to about $10^6$. Since WTA may be distributed throughout and on both sides of the PG layer, let us concentrate on the LTA which are less abundant but are all expected to be in the periplasmic space, with about $1.5 \times 10^5$ chains per cell. A single pneumococcal cell could be approximated by a sphere with a diameter of 1 μm and surface area of $3 \times 10^{-12}$ m$^2$. Pneumococcus exists mostly as diplococci, so let us assume an average surface area per cell of $5 \times 10^{-12}$ m$^2$. The surface density of LTA would therefore be $3.3 \times 10^{16}$ molecule·m$^{-2}$ or one LTA chain per 30 nm$^2$. This density value is lower than that of one chain per 5 nm$^2$ calculated for LTA from *Staphylococcus aureus* based on the ratio of LTA to membrane lipids (*Erickson, 2021*). Although both density values are rough estimates, they likely require volume exclusion to accommodate TA chains at the membrane surface and are probably compatible with the polymer brush model.

The morphogenesis and cell division of Gram-positive bacteria rely on the carefully orchestrated assembly of the cell envelope, which includes the membrane, the PG layer and the TA. To gain a better understanding of the incorporation of TA into the cell wall, TA were metabolically labeled with aCho that allowed the subsequent grafting of fluorescent probes. Cells labeled during short incubation pulses could be observed by dSTORM, revealing the regions of TA insertion in unprecedented details. In parallel, labeled TA, after short pulses or continuous growth in labeling media, could be analyzed by gel electrophoresis after cell wall digestion by PG hydrolases. As expected from an earlier study with conventional microscopy (*Bonnet et al., 2018a*), TA are incorporated in the cell wall at mid-cell (*Figure 3*, *Figure 3—figure supplement 1* and *Figure 3—figure supplement 2*). Patterns after a 5-min labeling pulse are very similar to those obtained when the PG is pulse-labeled (*Trouve et al., 2021b*) and do not differ in the absence of the glycosyltransferase TacL. In contrast, the absence of the phosphotransferase LytR markedly reduces the fluorescence signal. These observations are consistent with the role attributed to TacL of anchoring the less abundant LTA, and to LytR of attaching the more abundant WTA (*Heß et al., 2017*; *Minhas et al., 2023*). The electrophoretic analysis of labeled TA confirmed that WTA are less abundant in the Δ*lytR* strain (*Figure 6*). The amount of LTA appeared comparable in WT and Δ*lytR* cells, but LTA seem to contribute little to the dSTORM localization patterns, since cell wall sacculi that lacks LTA show patterns identical to that of whole cells (*Figure 3—figure supplement 2*).

The notable difference with TA compared to PG is the presence of some heterogenous surface localization after a labeling pulse. This surface signal is not due to non-specific binding of the fluorophore to the surface, since it is not observed in labeling studies of the PG (*Trouve et al., 2021b*). Since the same level of heterogenous surface labeling was observed with the three strains investigated, the most likely reason is some non-specific adsorption of aCho at the cell surface. It is also possible that

some LTA or membrane-bound precursors, which are not covalently bound to the cell wall, can diffuse from the division site across the plasma membrane during the pulse and prior the fixation. Finally, it is possible that hitherto unidentified choline-containing compounds at the cell surface are also labeled.

When imaging was performed after a chase time during which cells are allowed to grow in the absence of probes following the labeling pulse, patterns of labeled TA in Δ*tacL* and Δ*lytR* cells appeared different from those produced when the PG was labeled (*Figure 3*). In WT cells, in contrast, TA and PG labeling patterns were similar throughout the chase time. Although the appearance of pulse labeled Δ*tacL* cells resemble that of WT cells, the labeling patterns are strikingly different after a chase period. In Δ*tacL* cells, all the cell wall generated during the chase is labeled, resulting in a wide contiguous fluorescent region, whereas it is not labeled in WT cells, producing separated bands. It is most likely that in WT cells, production of WTA and LTA are competing pathways as the two polymers share the same composition, that rapidly consumes the undecaprenyl-linked polymerized TA precursors, whereas these precursors accumulate in the Δ*tacL* strain (*Figure 6B–E*). The slower turnover of the undecaprenyl-linked TA in the Δ*tacL* strain allows the cell wall incorporation of labeled WTA during the chase, several minutes after the aCho has been removed from the medium. This phenomenon could be observed in an electrophoretic analysis of a pulse–chase labeling experiment of the Δ*tacL* strain, where membrane bound TA precursors disappear over a 15-min course (*Figure 6B*). Patterns are difficult to discern in the Δ*lytR* strain due to the weak labeling signal, however, they appear to resemble those of Δ*tacL* cells.

In streptococci of the *mitis* group, TA serve as moorings for the attachment of a variety of surface proteins termed choline-binding proteins, because the interaction occurs between the choline residues of TA and protein domains made of repeated choline-binding motifs. The functions of these proteins are varied, playing roles in virulence, competence, or cell wall metabolism (*Maestro and Sanz, 2016*). The localization of some choline-binding proteins can be restricted, such as that of the autolysin LytA, which is found at mid-cell to hydrolyze the PG to promote lysis, or the other cell wall hydrolase LytB that is enriched at the new poles to complete cell separation (*Bonnet et al., 2018a*; *De Las Rivas et al., 2002*). Since the choline-binding domains of these proteins participate in their localization (*Bonnet et al., 2018a*; *Martínez-Caballero et al., 2023*), it probably requires that TA at different places of the cell surface display some compositional or size differences, that could arise from some maturation process. To investigate this possibility, we analyzed metabolically labeled TA at different stages of cell culture (*Figure 5*, *Figure 5—figure supplement 1*). Interestingly, longer LTA are present when the culture enters its stationary phase. One can speculate that the same may be true with WTA, although this could not be determined in our experiment. Surprisingly, when pulse labeling is performed at the onset of stationary phase, the LTA that are synthesized during that phase of the culture do not differ from those produced earlier during the exponential phase, despite the observation of longer LTA in late phase culture (*Figure 5A, B*, *Figure 5—figure supplement 1*). How could these long LTA arise? Since their length distribution is bi-modal and the longer (12–16 units) are about twice the size of the short ones (5–7 units), it is tempting to imagine that they may arise from the coupling of pre-existing TA.

The finding that undecaprenyl-attached TA accumulate in the Δ*tacL* strain to a level that is lower, but comparable to that of LTA in a WT strain raises the question of whether these precursors can fulfill the function of mature LTA, at least partially. The observation by CEMOVIS that the cell envelope architecture is altered in Δ*tacL* cells, with a thinner periplasmic space devoid of granular layer (*Figure 1*), indicates that if some functional replacement occurs, it is not complete. It has been proposed previously that the physical properties of the undecaprenyl chain underlie the localization of the PG precursor lipid II at the sites of cell wall assembly, which occurs at sites of maximal membrane curvature (*Calvez et al., 2019*). The same reasoning would restrict the presence, and thus the function, of the undecaprenyl-linked TA precursors to sites of cell wall assembly were they can be incorporated into WTA or true LTA. In the absence of TacL, undecaprenyl-linked TA accumulate to a greater extent, so that their turnover is slower than in a WT strain, resulting in the incorporation of labeled WTA during a chase after the labeling pulse (*Figure 3*).

The propensity of *S. pneumoniae* membrane to sediment at low centrifugal speed, discussed in the SI, is not without experimental consequences in the study of TA. A number of recent studies have investigated the possible role of changes in the relative proportion of WTA and LTA in physiological processes such as autolysis or competence (*Minhas et al., 2023*; *Flores-Kim et al., 2019*; *Flores-Kim*

*et al., 2022*). However, it is likely that most of the LTA-containing membranes were unknowingly discarded in these experiments during a clarification step of cell lysates, suggesting that the reported ratio of LTA to WTA were underestimated. The identification of undecaprenyl-linked TA accumulated in the absence of TacL explains the troubling observation of LTA-like signal in the ΔtacL strains (*Flores-Kim et al., 2019*; *Flores-Kim et al., 2022*; *Wu et al., 2014*). The metabolic labeling and electrophoretic analysis methods reported here will certainly help to further study multiple aspects of the biology of TA in *S. pneumoniae*, such as the relationship with the capsule and the *O*-acetylation of the PG, or the specific localization and activity of CBPs.

## Materials and methods

### Key resources table

| Reagent type (species) or resource | Designation | Source or reference | Identifiers | Additional information |
|---|---|---|---|---|
| Strain, strain background (*S. pneumoniae*) | WT | Gift from J.-P. Claverys, Toulouse, France | sspCM246 | R800, *rpsL1*, Δ*cps*, *adr*⁺, StrR |
| Strain, strain background (*S. pneumoniae*) | Δ*tacL* | This study | sspCM514 | R800, *rpsL1*, Δ*tacL*, Δ*cps*, *adr*⁺, StrR |
| Strain, strain background (*S. pneumoniae*) | Δ*lytR* | This study | sspCM493 | R800, *rpsL1*, Δ*lytR*, Δ*cps*, *adr*⁺, StrR |

### Bacterial strains and culture conditions

*S. pneumoniae* strains are listed in the Key resource table. Glycerol stocks of WT, Δ*lytA*, Δ*tacL*, and Δ*lytR* cells were used to inoculate Bacto BHI broth (BD), C-medium (*Trouve et al., 2021a*). Cultures were grown at 37°C in a static incubator with a 5% $CO_2$ atmosphere. Two successive dilutions of the cultures were performed to ensure that steady-state growth was reached. To characterize growth phases, cells were grown in 2.5 ml of BHI or C-medium in 24-well plates sealed with a transparent film. Cultures were inoculated at an $OD_{600}$ of 0.03 (or 0.12 for the Δ*lytR* strain in C-medium). In a BMG Fluostar Omega plate reader at 37°C, turbidity was measured at 595 nm every 20 min after 5 s agitation.

### High-pressure freezing

Cells at an $OD_{600}$ of 0.2–0.3 were collected by centrifugation. Pellets were supplemented with or without 20% dextran in phosphate-buffered saline (PBS). Pellets supplemented with dextran were drawn into copper tubes. Pellets without dextran were dispensed in 3-mm type A gold/copper platelets, covered with the flat side of a 3-mm type-B aluminum platelet (Leica Microsystems). The samples were vitrified by high-pressure freezing using an HPM100 system (Leica Microsystems) in which cells were subjected to a pressure of 210 MPa at −196°C.

### CEMOVIS and CETOVIS

Copper tubes were trimmed at <−140°C in an EM UC7 Cryo-ultramicrotome equipped with a micromanipulator (Leica Microsystems). Ultrathin sections were produced with a 35° diamond cryo-knife (Diatome) at a nominal thickness of 70 nm and at a nominal cutting feed of 50 mm·s⁻¹. Sections were transferred onto Quantifoil carbon-covered 200-mesh copper grids using the micromanipulator and an antistatic device for favoring the attachment of the ribbon to the grid. Grids were transferred to a Gatan cryoholder kept at −170°C and inserted into a TF20 cryo-electron microscope (FEI) equipped with a tungsten field emission gun. The accelerating voltage was 200 kV. Specimens were irradiated with a low electron dose. Images were recorded with a CMOS CETA camera at a nominal magnification ranging from ×5000 to ×29,000 (pixel size 8.1–3.7 Å). No peculiar image processing was performed on the micrograph. For CETOVIS, tilt series were acquired from −50° to +50° with an increment of 2.5° according to the dose symmetric Hagen scheme. Images were acquired at a defocus of −10 µm at nominal magnification ×29,000 (pixel size 3.7 Å). Tilt series were aligned, and tomogram reconstructed using the IMOD software package (*Mastronarde, 2006*), using the patch tracking method, followed by weighted back projection and SIRT like filter for purposes of representation.

## Freeze-substitution and ultramicrotomy at room temperature

The freeze-substitution and ultramicrotomy protocols were derived from *Bauda et al., 2024*. Briefly, following high-pressure freezing, the vitrified pellets were freeze-substituted at –90°C for 80 hr in acetone supplemented with 2% $OsO_4$ (AFS2; Leica Microsystems). The samples temperature was increased slowly to –60°C (2°C·hr⁻¹). The temperature was further raised to –30°C (2°C·hr⁻¹) after 8–12 hr, and finally to 0°C within 1 hr. The temperature was decreased to –30°C within 30 min, and rinsed in pure acetone four times. The samples were infiltrated with progressively higher concentrations of resin (Embed812, EMS) in acetone, while the temperature was gradually increased to 20°C. Pure resin was supplemented at room temperature. Following polymerization at 60°C for 48 hr, 70 nm thick sections were produced using a Leica UC7 ultramicrotome and placed onto formvar carbon-coated 200-mesh copper grids (Agar Scientific). Uranyl acetate (2% during 5 min) was used to stain the sections, followed by a 5-min incubation in lead citrate. After rinsing in water, the sections were imaged using a Tecnai G2 spirit BioTwin (FEI) microscope operating at 120 kV, at nominal magnifications of 11,000 to ×23,000 (pixel size of 5.6–2.8 Å), with an Orius SC1000B CCD camera (Gatan).

## Conventional and super-resolved fluorescence microscopy

Cultures in exponential phase (25 ml, $OD_{600}$ of 0.3) were centrifuged at 3220 × *g* at room temperature for 15 min. The bacterial pellets were resuspended in 1/25th original volume of pre-warmed BHI supplemented with either 2 mM azido-D-Ala-D-Ala for PG labeling or 1.5 mM aCho for labeling of the TA, and incubated for 5 min at 37°C water bath (the 'pulse' period).

Cells were then pelleted (10,000 × *g*, 1 min) and washed with 1 ml BHI before being resuspended in 20 ml of pre-warmed BHI without probe. Immediately, 2 ml of this resuspension were pelleted (10,000 × *g*, 1 min) to prepare 'pulse-only' samples. The rest of the bacterial culture was allowed to continue growing at 37°C ('chase' period). At 5, 15, and 35 min into the 'chase' period, 2 ml were pelleted as described earlier. The pellets were resuspended in PBS containing 2% paraformaldehyde for overnight fixation at 4°C.

For the preparation of sacculi, the 'pulse' period was conducted as above. After cells were washed and resuspended in fresh medium, they were allowed to grow further for 15 min. The chased bacterial cultures (19 ml) were centrifuged (3220 × *g*, 5 min) and the pellets were washed with 1 ml of 10% SDS (10,000 × *g*, 1 min). Cells were then resuspended in 1 ml of 10% SDS and incubated at 100°C for 1 hr, vortexed every 15 min. Sacculi were washed twice with 500 µl of 80 mM Tris-HCl pH 7 and incubated in 500 µl of a solution containing 20 µg·ml⁻¹ DNAse and 20 µg·ml⁻¹ RNAse, 20 mM $MgCl_2$ and 80 mM Tris-HCl pH 7 at 37°C with agitation for 1 hr, prior to addition of Proteinase K (final concentration 200 µg·ml⁻¹), 2 mM $CaCl_2$, 80 mM Tris-HCl pH 7, and further incubation overnight. Finally, sacculi were washed once with 200 µl of water.

For coupling of fluorophores, cells or sacculi were pelleted (10,000 × *g*, 1 min) and resuspended into 50 µl (for fixed cells) or 150 µl (for sacculi) of PBS containing 30 µM DBCO-AF647 (JenaBiosciences). After 45 min at room temperature, samples were washed three times with 300 µl PBS. For dSTORM imaging, fluorophore-labeled cells or sacculi were resuspended in a solution containing 100 mM β-mercaptoelthylamine and an oxygen-depleting system consisting of a GLOX enzyme mix (40 µg·ml⁻¹ catalase, 0.5 mg·ml⁻¹ glucose oxidase) in 75 mM Tris-HCl pH 8, 25 mM NaCl, and 10% glucose.

Samples were mounted between a microscopy slide and a high-precision coverslip pre-treated with UV light for 20 min. To coax the cells or sacculi into lying on the slide along their longitudinal axis, a heavy weight was applied on top of the cover slip as described previously (*Trouve et al., 2021a*). The edges of the cover slip were sealed with colorless nail polish.

Observations were made with an Abbelight SAFe 360 commercial STORM microscope. For fluorescence signal acquisition, the field was first excited with a 640-nm scanning laser at low power (1%) to capture a low-resolution conventional fluorescence image of the AF647 signal. The laser intensity was then steadily raised to 100%. ASTER technology, implemented in the microscope, provides for a homogeneous sample excitation. In our experiments, the gaussian beam at the power density of 10 kW·cm⁻² (average, FWHM 50 µm, power 200 mW, measured at the sample) scans the region of interest of 160² µm² resulting in 800 W·cm⁻² of average effective power density. AF647 dye molecules went through cycles of fluorescent state and dark state. When most AF647 fluorophores had displayed this transition estimated visually by the frequency and density of fluorescence blinking, a set

of 15,000 consecutive frames were taken with 50-ms exposure time. A bright-field image of the field was captured afterwards.

The 15,000 frame sets were processed with the FIJI plugin ThunderSTORM, which localizes the center of a Gaussian function fitted to each individual fluorescence signal and returns a data table containing the localization coordinates of all labeled molecules. Correction of drifts occurring during each set of acquisition and reconstruction of the final super-resolution images were performed in the same software.

## TA labeling, cell lysis, and fractionation for electrophoretic analysis

TA were labeled by growing cells WT or mutant strains in C-medium in the presence of 200 µM aCho to an $OD_{600}$ of 0.5. Cells were harvested, washed and resuspended to an $OD_{600}$ of 15 in 50 mM Tris-HCl pH 8, 150 mM NaCl, 1 mM $MgCl_2$ containing 0.36 mg·ml$^{-1}$ lysozyme, 0.32 mg·ml$^{-1}$ mutanolysin, 0.36 mg·ml$^{-1}$ RNase, 0.36 mg·ml$^{-1}$ DNase, and 25 µM DBCO-AF488. After lysis overnight on wheel at room temperature cells were centrifuged for 2 min at 1250, 5000, or 20,000 × g. The pellets were resuspended in the same volume. An aliquot of the 20,000×g supernatant was incubated at 37°C for 2 hr with 10 kU of calf intestinal alkaline phosphatase.

Samples were then analyzed by electrophoresis on polyacrylamide gel (12%, acrylamide:bis-acrylamide 29:1) in the presence of 1% SDS with a Tris/tricine buffer system. The gels were imaged by trans-illumination with UV light without saturation on a BioRad Chemidoc imager.

## Lipid analysis

WT cells were grown to an $OD_{600}$ of 0.5 in BHI and harvested by centrifugation for 15 min at 3220 × g. Cells were washed twice with 1 ml 50 mM Tris-HCl pH 8, 150 mM NaCl, 1 mM $MgCl_2$ and resuspended in 1 ml of the same solution containing 0.36 mg·ml$^{-1}$ lysozyme, 0.32 mg·ml$^{-1}$ mutanolysin, 0.36 mg·ml$^{-1}$ RNase, and 0.36 mg·ml$^{-1}$ DNase. After overnight incubation at room temperature. Half of the sample was centrifuged for 2 min at 20,000 × g, and the pellet was resuspended in the initial volume of the same buffer. The pellet, supernatant, and whole 500 µl fractions were each extracted twice with 500 µl $CHCl_3$. The organic fractions were dried and redissolved in 40 µl $CHCl_3$/methanol (80:20) prior to analysis by thin layer chromatography on 0.2 mm silica plates (Macherey-Nagel Xtra SIL G) developed with $CHCl_3$/methanol/acetic acid (80:15:8) and stained with iodine vapour.

## Negative stain electron microscopy of pellets

Negative Stain-Mica-carbon Flotation Technique (MFT)-Valentine procedure was applied (***Valentine et al., 1968***). Samples were absorbed to the clean side of a carbon film on mica, stained and transferred to a 400-mesh copper grid. The images were taken under low dose conditions (<10 e$^-$·Å$^{-2}$) with defocus values between 1.2 and 2.5 µm on a Tecnai 12 LaB6 electron microscope at 120 kV accelerating voltage using CCD Camera Gatan Orius 1000. The stain was Sodium Silico Tungstate $Na_4O_{40}SiW_{12}$ at 1% in distilled water (pH 7–7.5).

## Isolation of membrane-bound TA for NMR analysis

WT cells grown in BHI to $OD_{600}$ 0.8 were harvested by centrifugation (20 min, 4500 × g) and resuspended in 10 ml of 50 mM Tris-HCl pH 8, 150 mM NaCl, 5 mM $MgCl_2$ with 10 µg·ml$^{-1}$ each of RNAse and DNAse, 9 µg·ml$^{-1}$ mutanolysin, 50 µg·ml$^{-1}$ lysozyme. After overnight incubation on a rotating wheel at 20°C, the lysate was centrifuged for 10 min at 15,000 × g. The pellet was washed thrice with 5 ml water and thrice with 3 ml 100 mM ammonium acetate pH 4.2. The pellet was resuspended in 2.5 ml of the same buffer and 2.5 ml of water-saturated butanol. After vigorous mixing for 30 min, phases were separated by centrifugation at 3220 × g for 10 min. The butanol phase was reextracted twice with 2.5 ml water. The aqueous phases were combined and extracted with 8 ml and then 2 ml $CHCl_3$. The resulting combined chloroform phases was reextracted with 4 ml water. All water phases were combined and lyophilized. One half of the sample was dissolved in 200 µl of $D_2O$ for NMR analysis. The other half of the sample was dissolved in 500 µl hydrazine 64% and incubated at room temperature for 4 hr prior to addition of 500 µl acetone. The sample was dried under nitrogen flow, dissolved in 500 µl acetone and dried again. After dissolution in 200 µl $D_2O$ and removal of insoluble material by centrifugation at 20,000 × g for 10 min, the hydrazinolized sample was analyzed by NMR.

## NMR analysis

NMR experiments were recorded on 600 and 700 MHz spectrometers equipped with a 5-mm TCI probe at 50°C. $^{13}$C-$^{1}$H HSQC (constant time) and 2D $^{13}$C-edited $^{1}$H-$^{1}$H total correlated spectroscopy (TOCSY) were used to confirm $^{1}$H-$^{13}$C assignment. $^{31}$P 1D experiment was recorded with $^{1}$H decoupling during acquisition and a recycling delay of 1 s. Data were processed with TopSpin 3.5 (Bruker) and analyzed with CcpNmr 2.5.

## Titration of cellular TA

A preculture of WT cells in BHI was washed twice with C-medium without choline prior to inoculating a C-medium culture containing 200 µM aCho at an $OD_{600}$ of 0.06. After incubation at 37°C, cells were harvested when the $OD_{600}$ reached 0.265, that is after more than two generations in C-medium.

Cells from 10 ml culture were resuspended in 160 µl of 50 mM Tris-HCl pH 8, 150 mM NaCl, 8 mM $MgCl_2$ containing 0.4 mg·ml$^{-1}$ lysozyme, 0.3 mg·ml$^{-1}$ mutanolysin, 0.3 mg·ml$^{-1}$ recombinant pneumo-coccal LytA, and 0.4 mg·ml$^{-1}$ each of RNase and DNase. The suspension was left on wheel overnight at room temperature to allow complete lysis.

Considering that a cell suspension at an optical $OD_{600}$ of 1 contains about $3.33 \times 10^8$ cells ml$^{-1}$, the cell concentration of the lysate was about $5.52 \times 10^8$ ml$^{-1}$. The concentration of the stock solution of DBCO-AF488 was determined by its absorbance at 494 nm using $\varepsilon = 73,000$ M$^{-1}$·cm$^{-1}$ to be $93 \pm 5$ µM (sd).

The cell lysate and dilutions thereof were mixed with an equal volume of DBCO-AF488 in 50 mM Tris-HCl pH 8, 150 mM NaCl to final concentrations of $3.7 \pm 0.2$ and $1.9 \pm 0.1$ µM of DBCO-AF488 and a maximum of $2.8 \times 10^8$ cells·ml$^{-1}$. After 24 hr of incubation at room temperature, 1 µl of 100 mM aCho were added to block unreacted DBCO-AF488 and incubation was continued for 2 hr.

Samples were then analyzed by electrophoresis on polyacrylamide gel (17%, acrylamide:bis-acrylamide 29:1) in the presence of 1% SDS with a Tris/tricine buffer system. The gels were imaged by trans-illumination with UV light without saturation on a Bio-Rad Chemidoc imager. Band intensities were quantified using ImageJ.

## TA characterization at different stages of cell culture

WT and Δ*lytA* cells were grown for over two generations in chemically defined medium (C-medium) devoid of choline but supplemented with 200 µM aCho over two generations. Cells were harvested during the exponential phase of growth at an $OD_{600}$ of 0.38, at the onset of the stationary phase at the $OD_{600}$ of 0.75, and when the WT cells started to lyse ($OD_{600}$ 0.46). TA were labeled and cell were lysed, LTA and WTA were separated by centrifugation and analyzed by electrophoresis as described above.

## TA pulse and pulse–chase labeling

Exponentially growing WT cells in C-medium were harvested and resuspended at an $OD_{600}$ of 8 in the same medium containing 200 µM aCho and no choline. After 0, 2, 5, and 10 min, 150 µl aliquots were withdrawn, immediately centrifuged for 30 s at $10,000 \times g$ and resuspended in the same volume of 50 mM Tris-HCl pH 8, 150 mM NaCl. The washing procedure was repeated once prior to lysis, secondary fluorescent click-labeling and fractionation as above. Pellets containing LTA were resuspended in one-eighth volume for concentration. TA were analyzed by gel electrophoresis as above.

## Hydrolysis of sedimented TA from Δ*tacL* cells

Three procedures were tested. First, pellets of labeled lysates of WT or Δ*tacL* cells were resuspended in solutions of 150 mM NaCl buffered with either 50 mM Tris-HCl pH 8 or ammonium acetate pH 4.2, and incubated for 3 hr at room temperature or 99°C.

Second, the LTA isolation procedure of Hess and colleagues (*Heß et al., 2017*) was applied. Labeled pellets resulting from 10 ml of culture of Δ*tacL* or 7 ml of WT cells were dissolved in 200 µl of 50 mM sodium citrate pH 4.7, 4% SDS and incubated at 100°C for 45 min. After withdrawal of a 20-µl aliquot for analysis, the solution was lyophilized. The solid was washed four times with ethanol and resuspended in 180 µl of 50 mM sodium citrate pH 4.7. A second 20 µl aliquot was withdrawn, and the solution was extracted with 150 µl water-saturated butanol. The organic phase was back extracted with 100 µl water. The aqueous phases were combined, lyophilized, redissolved in 160 µl of 50 mM

sodium citrate pH 4.7, and a 20-µl aliquot was taken for analysis. To the 20 µl aliquots, 2 µl of 3 M Tris-HCl pH 8.45 were added prior to analysis by electrophoresis.

Third, recombinant colicin M from *P. aeruginosa* (generous gift from T. Touzé) prepared as described previously (*Barreteau et al., 2012*), was added at various concentrations to resuspended pellets and the mixtures were incubated overnight at room temperature.

## Notes on the fractionation of LTA and WTA

Although sedimentation of LTA was observed at relative centrifugal force as low as 2000 × *g*, the pelleting was not always complete, even after prolonged centrifugation at 20,000 × *g*. Incomplete cell lysis cannot account for the incomplete pelleting of LTA, since intact cells or large fragments would be expected to add to the pellet material. We found out that the quality of the separation of LTA from solubilized WTA by sedimentation depended on the amount of recombinant LytA used during the lysis (*Figure 4—figure supplement 1A*). This parameter was difficult to adjust since the exactly adequate amount appeared to depend on the particulars of the experiment (strain, cell density). We think that the multivalent choline-binding domain of LytA crosslinks LTA bound to membrane vesicles, leading to the formation of large aggregates that sediment more easily. In support of this explanation, the presence of recombinant LytA also impacts the migration of the solubilized TA, causing species with lower electrophoretic mobilities to be more abundant at the top of the gels.

The fractionation of LTA and WTA with the procedure used previously (*Flores-Kim et al., 2019*; *Flores-Kim et al., 2022*), which involves the preparation of spheroplasts in high concentration of sucrose, was compared to the low-speed centrifugation following direct enzymatic lysis (*Figure 4—figure supplement 1B*). Taking advantage of the fluorescent labeling of TA, we found that most WTA and LTA were sedimented at 5000 × *g*, but some LTA were further pelleted at 20,000 and 100,000×*g*, whereas some WTA remained in the ultracentrifugation supernatant. The formation of spheroplasts appeared be complete after 1 h of cell wall digestion as checked by phase-contrast microscopy. However, intact cells and/or large cell wall fragments were still present after hypotonic lysis for 30 min since most of the WTA were pelleted at low speed. It is likely that the extent of lysis depends on the quality of the enzyme stocks that are used and may be difficult to replicate in different laboratories. Also, large membrane fragments or intact spheroplasts may sediment more easily than smaller membrane fragments. Thus, the amount of LTA-containing membranes sedimented at the different centrifugal speeds may depend on the specific handling of the samples (e.g. more or less vigorous resuspensions).

Indeed, when cells, mechanically broken using a microfluidizer, were fractionated at different centrifugal speeds (*Figure 4—figure supplement 1C*), most of the membranes were found in the pellet sedimented at 100,000 × *g*. This different behavior indicates that the mechanical processing impacts the sedimentation properties of membranes, probably by changing the size of the fragments. Since the large minimal volume of culture compatible with the microfluidic equipment precluded labeling of the TA, the membranes were monitored by Western blot using an antibody against the membrane protein PBP2x. A fraction of the same culture was treated by direct enzymatic lysis and low-speed centrifugation. In this case, no PBP2x was detected, presumably due to degradation by proteases present in the commercial mutanolysin, even in the presence of protease inhibitors.

## Notes on the electrophoresis

Several points should be made regarding the electrophoretic system. We used gels with acrylamide concentrations of either 12% or 17%. Although the acrylamide concentration had little consequence on the separation of WTA and LTA, we observed that the migration of small soluble compounds was greatly affected (*Figure 4—figure supplement 2A, B*). Thus, the phospho-choline-triazole-AF488 migrates a lesser distance than the choline-triazole-AF488 or the DBCO-AF488 in 12% acrylamide gels, whereas the opposite occurs in 17% gels. The separation of the choline-triazole-AF488 and the DBCO-AF488 is not sufficient in 17% gels.

Also, we found that thiol reducing agents such as DTT routinely included when loading the gels do react with DBCO-AF488 to yield a mixture of species, denoted with * in *Figure 4—figure supplement 2A, B*. The extent of this reaction appeared to depend on the length and conditions of the storage of the samples prior to analysis and therefore was not controlled.

In some cases, such as in *Figure 6A, B or E*, one or multiple additional weak but sharp bands of fluorescently labeled material appear on the gels above the LTA bands. These bands can be correlated to Coomassie-stained protein bands (*Figure 4—figure supplement 3D–F*) and they are likely products of the reaction of DBCO-AF488 with protein thiols or primary amines. As stated above for the modification of DBCO-AF488, this side reaction was not controlled and not always present. A weak non-specific fluorescent labeling of cellular proteins by DBCO-AF488, which was not a problem when TA were labeled during the whole culture, contributed to a significant smear blurring the TA bands when labeling was performed only during a short pulse time such as in *Figure 5C*.

## Acknowledgements

We thank D Fenel from the IBS for performing the negative stain TEM, and T Touzé from the Université Paris Saclay for the generous gift of purified colicin M. We thank O Glushonkov, J-P Kleman for advice and support regarding dSTORM; G Schoehn for electron microscopy. We thank H Erickson from Duke University for sharing thoughts about TA and periplasmic space. Support for this work comes from the Agence Nationale de la Recherche (ANR-23-CE11-0029 to CeM, ANR-19-CE15-0011 to CG, and ANR-19-CE07-0035 to YSW). EB received funding from GRAL, a program from the Chemistry Biology Health (CBH) Graduate School of University Grenoble Alpes (ANR-17-EURE-0003) and the Ecole Doctorale Chimie et Sciences du Vivant of the University Grenoble Alpes. IBS acknowledges integration into the Interdisciplinary Research Institute of Grenoble (IRIG, CEA). This work used the platforms of the Grenoble Instruct-ERIC center (ISBG; UAR 3518 CNRS-CEA-UGA-EMBL) within the Grenoble Partnership for Structural Biology (PSB), supported by FRISBI (ANR-10-INBS-0005-02) and Labex GRAL and ARCANE, financed within the University Grenoble Alpes graduate school (Ecoles Universitaires de Recherche) CBH-EUR-GS (ANR-17-EURE-0003). The IBS/ISBG electron microscope facility is supported by the Auvergne-Rhône-Alpes Region, the Fondation Recherche Médicale (FRM), the fonds FEDER, and the GIS-Infrastructures en Biologie Santé et Agronomie (IBISA).

## Additional information

### Funding

| Funder | Grant reference number | Author |
| --- | --- | --- |
| Agence Nationale de la Recherche | ANR-23-CE11-0029 | Cecile Morlot |
| Agence Nationale de la Recherche | ANR-19-CE15-0011 | Christophe Grangeasse |
| Agence Nationale de la Recherche | ANR-19-CE07-0035 | Yung-Sing Wong |
| LabEx Gral | ANR-17-EURE-0003 | Elda Bauda |
| FRISBI | ANR-10-INBS-0005-02 | Cecile Morlot |
| LABoratoires d'EXcellence ARCANE | ANR-17-EURE-0003 | Cecile Morlot |

The funders had no role in study design, data collection, and interpretation, or the decision to submit the work for publication.

### Author contributions

Mai Nguyen, Investigation, Visualization, Writing – original draft, Writing – review and editing; Elda Bauda, Investigation, Visualization, Methodology, Writing – original draft, Writing – review and editing; Célia Boyat, Céline Freton, Anne Chouquet, Investigation; Cédric Laguri, Investigation, Writing – original draft; Benoit Gallet, Morgane Baudoin, Investigation, Methodology; Yung-Sing Wong, Funding acquisition, Methodology; Christophe Grangeasse, Supervision, Funding acquisition; Christine Moriscot, Supervision, Methodology; Claire Durmort, Supervision, Investigation, Methodology; André Zapun, Conceptualization, Formal analysis, Supervision, Visualization, Writing – original draft, Project

administration, Writing – review and editing; Cecile Morlot, Conceptualization, Supervision, Funding acquisition, Writing – original draft, Project administration, Writing – review and editing

#### Author ORCIDs
Elda Bauda http://orcid.org/0000-0003-3848-6141
Cédric Laguri https://orcid.org/0000-0002-5429-6914
Benoit Gallet http://orcid.org/0000-0001-8758-7681
Christophe Grangeasse http://orcid.org/0000-0002-5484-4589
Claire Durmort http://orcid.org/0000-0001-5362-5061
André Zapun https://orcid.org/0000-0001-8953-4399
Cecile Morlot http://orcid.org/0000-0002-9295-1035

Reviewer #1 (Public review): https://doi.org/10.7554/eLife.105132.3.sa1
Reviewer #2 (Public review): https://doi.org/10.7554/eLife.105132.3.sa2
Author response https://doi.org/10.7554/eLife.105132.3.sa3

---

## Additional files

#### Supplementary files
MDAR checklist

#### Data availability
Unprocessed EM and optical microscopy images are available on Zenodo at https://doi.org/10.5281/zenodo.15111725 and https://doi.org/10.5281/zenodo.15111761, respectively. Unprocessed gel images are available at https://doi.org/10.5281/zenodo.15111783. Measurements for Table 1 and corresponding EM images are available at https://doi.org/10.5281/zenodo.15119160.

The following previously published datasets were used:

| Author(s) | Year | Dataset title | Dataset URL | Database and Identifier |
|---|---|---|---|---|
| Zapun A, Morlot C, Bauda E | 2025 | Figure 2-Source Data | https://doi.org/10.5281/zenodo.15111725 | Zenodo, 10.5281/zenodo.15111725 |
| Zapun A, Morlot C, Nguyen M | 2025 | Figure 3-Source Data | https://doi.org/10.5281/zenodo.15111761 | Zenodo, 10.5281/zenodo.15111761 |
| Zapun A, Morlot C, Bauda E, Nguyen M | 2025 | Figures 4, 5 & 6-Source Data | https://doi.org/10.5281/zenodo.15174083 | Zenodo, 10.5281/zenodo.15174083 |
| Morlot C, Zapun A, Bauda E, Nguyen M | 2025 | Table 1-Source Data | https://doi.org/10.5281/zenodo.15119160 | Zenodo, 10.5281/zenodo.15119160 |

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
