## [Editor Report · eLife Assessment]

The bacterial cell wall is crucial to maintain viability. It has previously been suggested that Gram positive bacteria have a periplasmic region between the cell membrane and peptidoglycan cell wall that this is maintained by the presence of teichoic acids. In this **valuable** study, Nguyen et al. make clever use of electron microscopy and metabolic labelling to interrogate the role of teichoic acids in supporting the maintenance of the periplasmic region in Streptococcus pneumoniae. The findings are **solid** and close some crucial knowledge gaps whilst providing novel tools to further interrogate discrepancies in the field. This work will be of broad interest to microbiologists.

---

## [Referee Report · Reviewer #1 (Public review)]

The authors set out to analyse the roles of the teichoic acids of Streptococcus pneumoniae in supporting the maintenance of the periplasmic region. Previous work has proposed the periplasm to be present in Gram positive bacteria and here advanced electron microscopy approach was used. This also showed a likely role for both wall and lipo-teichoic acids in maintaining the periplasm. Next, the authors use a metabolic labelling approach to analyse the teichoic acids. This is a clear strength as this method cannot be used for most other well studied organisms. The labelling was coupled with super-resolution microscopy to be able to map the teichoic acids at the subcellular level and a series of gel separation experiments to unravel the nature of the teichoic acids and the contribution of genes previously proposed to be required for their display. The manuscript could be an important addition to the field but there are a number of technical issues which somewhat undermine the conclusions drawn at the moment. These are shown below and should be addressed. More minor points are covered in the private

Recommendations for Authors.

Weaknesses to be addressed:

(1) l. 144 Was there really only one sample that gave this resolution? Biological repeats of all experiments are required.

(2) Fig. 4A. Is the pellet recovered at "low" speeds not just some of the membrane that would sediment at this speed with or without LTA? Can a control be done using an integral membrane protein and Western Blot? Using the tacL mutant would show the behaviour of membranes alone.

(3) Fig. 4A. Using enzymatic digestion of the cell wall and then sedimentation will allow cell wall associated proteins (and other material) to become bound to the membranes and potentially effect sedimentation properties. This is what is in fact suggested by the authors (l. 1000, Fig. S6). In order to determine if the sedimentation properties observed are due to an artefact of the lysis conditions a physical breakage of the cells, using a French Press, should be carried out and then membranes purified by differential centrifugation. This is a standard, and well-established method (low-speed to remove debris and high-speed to sediment membranes) that has been used for S. pneumoniae over many years but would seem counter to the results in the current manuscript (for instance Hakenbeck, R. and Kohiyama, M. (1982), Purification of Penicillin-Binding Protein 3 from Streptococcus pneumoniae. European Journal of Biochemistry, 127: 231-236).

(4) l. 303-305. The authors suggest that the observed LTA-like bands disappear in a pulse chase experiment (Fig. 6B). What is the difference between this and Fig. 5B, where the bands do not disappear? Fig. 5C is the WT and was only pulse labelled for 5 min and so would one not expect the LTA-like bands to disappear as in 6B?

(5) Fig. 6B, l. 243-269 and l. 398-410. If, as stated, most of the LTA-like bands are actually precursor then how can the quantification of LTA stand as stated in the text? The "Titration of Cellular TA" section should be re-evaluated or removed? If you compare Fig. 6C WT extract incubated at RT and 110oC it seems like a large decrease in amount of material at the higher temperature. Thus, the WT has a lot of precursors in the membrane? This needs to be quantified.

(6) L. 339-351, Fig. 6A. A single lane on a gel is not very convincing as to the role of LytR. Here, and throughout the manuscript, wherever statements concerning levels of material are made, quantification needs to be done over appropriate numbers of repeats and with densitometry data shown in SI.

(7) 14. l. 385-391. Contrary to the statement in the text, the zwitterionic TA will have associated counterions that results in net neutrality. It will just have both -ve and +ve counterions in equal amounts (dependent on their valency), which doesn't matter if it is doing the job of balancing osmolarity (rather than charge).

Comments on revisions:

The resubmitted manuscript now contains new data and changes to the text.

The authors have largely covered my previous points in both sets of reviews (Public/Recommendations).

Public Review Points:

1 & 6: I still do not see a reproducibility statement as such, with details of the number of biological repeats etc.

2 & 3. Fig S7 seems to be quite telling. As predicted after physical breakage the membrane proteins sediment at high speed (rather than low speed). This presumably also means that the LTA comes down at high and not low speed. LTA was not measured due to cost of reagents. The Microfluidizer breaks the cells using a shear force and thus is unlikely to create very small membrane fragments. Thus, the sedimentation properties of membranes containing LTA are likely dependent on the way in which the cells are lysed. It is therefore worthwhile qualifying the statements on l. 35-36, 46-47 and 212 (as Ref 8 used mechanical breakage). This will give better direction to those in the field following up the findings.

It is also a little alarming that the mutanolysin is contaminated by protease and one hopes this does not affect any of the properties of the materials being analysed.

---

## [Referee Report · Reviewer #2 (Public review)]

The Gram-positive cell wall contains for a large part of TAs, and is essential for most bacteria. However, TA biosynthesis and regulation is highly understudied because of the difficulties in working with these molecules. This study closes some of our important knowledge gaps related to this and provides new and improved methods to study TAs. It also shows an interesting role for TAs in maintaining a 'periplasmic space' in Gram positives. Overall, this is an important piece of work. Future work will need to address the possible causal link between TAs and periplasmic space, for instance using complemented mutants and CEMOVIS. It will be interesting to see what happens with the periplasmic space in other mutants besides TA or also in strains with capsules/without capsules and in PG mutants, or in lafB (essential for production of another glycolipid) mutants. Overall, I support the publication of this revised work as it pioneers some new methods that will definitively move the field forward.

---

## [Author Response]

The following is the authors’ response to the original reviews

**Public Reviews:**

**Reviewer #1 (Public review):**
The authors set out to analyse the roles of the teichoic acids of Streptococcus pneumoniae in supporting the maintenance of the periplasmic region. Previous work has proposed the periplasm to be present in Gram positive bacteria and here advanced electron microscopy approach was used. This also showed a likely role for both wall and lipo-teichoic acids in maintaining the periplasm. Next, the authors use a metabolic labelling approach to analyse the teichoic acids. This is a clear strength as this method cannot be used for most other well studied organisms. The labelling was coupled with super-resolution microscopy to be able to map the teichoic acids at the subcellular level and a series of gel separation experiments to unravel the nature of the teichoic acids and the contribution of genes previously proposed to be required for their display. The manuscript could be an important addition to the field but there are a number of technical issues which somewhat undermine the conclusions drawn at the moment. These are shown below and should be addressed. More minor points are covered in the private Recommendations for Authors.Weaknesses to be addressed:(1) l. 144 Was there really only one sample that gave this resolution? Biological repeats of all experiments are required.

CEMOVIS is a very challenging method that is not amenable to numerous repeats. However, multiple images were recorded from at least two independent samples for each strain. Additional sample images are shown in a new Fig. S3.

CETOVIS is even more challenging (only two publications in Pubmed since 2015) and was performed on a single ultrathin section that, exceptionally, laid perfectly flat on the EM grid, allowing tomography data acquisition on ∆*tacL* cells. The reconstructed tomogram confirmed the absence of a granular layer in the depth of the section. Additionally, the numbering of Fig. S4A-B (previously misidentified as Fig. S2A-B) has been corrected in the text of V2.

(2) Fig. 4A. Is the pellet recovered at "low" speeds not just some of the membrane that would sediment at this speed with or without LTA? Can a control be done using an integral membrane protein and Western Blot? Using the tacL mutant would show the behaviour of membranes alone.

We think that the pellet is not just some of the membrane but most of it. In support of this view, the “low” speed pellets after enzymatic cell lysis contain not just some membrane lipids, but most of them (Fig. S10A). We therefore expect membrane proteins to be also present in this fraction. We performed a Western blot using antibodies against the membrane protein PBP2x (new Fig. S7C). Unfortunately, no signal was detected most likely due to protein degradation from contaminant proteases that we could trace to the purchased mutanolysin. The same sedimentation properties were observed with the ∆*tacL* strain as shown in Fig. 6A. However, in the ∆*tacL* strain the membrane pellet still contains membrane-bound TA precursors. It is therefore impossible to test definitely if pneumococcal membranes totally devoid of TA would sediment in the same way.

(3) Fig. 4A. Using enzymatic digestion of the cell wall and then sedimentation will allow cell wall associated proteins (and other material) to become bound to the membranes and potentially effect sedimentation properties. This is what is in fact suggested by the authors (l. 1000, Fig. S6). In order to determine if the sedimentation properties observed are due to an artefact of the lysis conditions a physical breakage of the cells, using a French Press, should be carried out and then membranes purified by differential centrifugation. This is a standard, and well-established method (low-speed to remove debris and high-speed to sediment membranes) that has been used for S. pneumoniae over many years but would seem counter to the results in the current manuscript (for instance Hakenbeck, R. and Kohiyama, M. (1982), Purification of Penicillin-Binding Protein 3 from Streptococcus pneumoniae. European Journal of Biochemistry, 127: 231-236).

Thank you for this suggestion. We have tested this hypothesis by breaking cells with a Microfluidizer followed by differential centrifugation. This experiment, which requires an important minimal volume, was performed with unlabeled cells (due to the cost of reagents) and assessed by Western blot using antibodies against the membrane protein PBP2x (new Fig. S7C). In this case, the majority of the membrane material was found in the high-speed pellet, as expected.

We also applied the spheroplast lysis procedure of Flores-Kim et al. to the labeled cells, and found that most of the labeled material sedimented at low speed (new Fig. S7B), as observed with our own procedure.

With these new results, the section on membrane density has been removed from the Supplementary Information. Instead, the fractionation is further discussed in terms of size of membrane fragments and presence of intact spheroplasts in the notes in Supplementary Information preceding Fig. S7.

(4) l. 303-305. The authors suggest that the observed LTA-like bands disappear in a pulse chase experiment (Fig. 6B). What is the difference between this and Fig. 5B, where the bands do not disappear? Fig. 5C is the WT and was only pulse labelled for 5 min and so would one not expect the LTA-like bands to disappear as in 6B?

Fig. 6B shows a pulse-chase experiment with strain ∆*tacL*, whereas Fig. 5C shows a similar experiment with the parental WT strain. The disappearance of the LTA-like band pattern with the ∆*tacL* strain (Fig. 6B), and their persistence in the WT strain (Fig. 5C), indicate that these bands are the undecaprenyl-linked TA in ∆*tacL* and proper LTA in the WT. A sentence has been added to better explain this point in V2.

Note that we have exchanged the previous Fig. 5C and Fig. S13B, so that the experiments of Fig. 5A and 5C are in the same medium, as suggested by Reviewer #2.

(5) Fig. 6B, l. 243-269 and l. 398-410. If, as stated, most of the LTA-like bands are actually precursor then how can the quantification of LTA stand as stated in the text? The "Titration of Cellular TA" section should be re-evaluated or removed? If you compare Fig. 6C WT extract incubated at RT and 110oC it seems like a large decrease in amount of material at the higher temperature. Thus, the WT has a lot of precursors in the membrane? This needs to be quantified.

Indeed, the quantification of the ratio of LTA and WTA in the WT strain rests on the assumption that the amount of membrane-linked polymerized TA precursors is negligible in this strain. This assumption is now stated in the Titration section. We think it is the case. The true LTA and TA precursors do not have exactly the same electrophoretic mobility, being shifted relative to each other by about half a ladder “step”. This difference is visible when samples are run in adjacent lanes on the same gel, as in the new Fig. 6C. The difference of migration was well documented in the original paper about the deletion of *tacL*, although *tacL* was known as *rafX* at that time, and the ladders were misidentified as WTA (Wu et al. 2014. A novel protein, RafX, is important for common cell wall polysaccharide biosynthesis in *Streptococcus pneumoniae*: implications for bacterial virulence. J Bacteriol. 196, 3324-34. doi: 10.1128/JB.01696-14). This reference was added in V2. The experiment in the new Fig. 6C was repeated to have all samples on the same gel and treated at a lower temperature. The minor effect on the amount of LTA when WT cells are heated at pH 4.2 may be due to the removal of some labeled phosphocholine. We have NMR evidence that the phosphocholine in position D is labile to acidic treatment of LTA, which may lack in some cases, as reported by Hess et al. (Nat Commun. 2017 Dec 12;8(1):2093. doi: 10.1038/s41467-017-01720-z).

(6) L. 339-351, Fig. 6A. A single lane on a gel is not very convincing as to the role of LytR. Here, and throughout the manuscript, wherever statements concerning levels of material are made, quantification needs to be done over appropriate numbers of repeats and with densitometry data shown in SI.

Yes indeed. Apart from the titration of TA in the WT strain, we haven’t yet carried out a thorough quantification of TA or LTA/WTA ratio in different strains and conditions, although we intend to do so in a follow-up study, using the novel opportunities offered by the method presented here.

However, to better substantiate our statement regarding the ∆*lytR* strain, we have quantified two experiments performed in C-medium with azido-choline, and two experiments of pulse labeling in BHI medium. The results are presented in the additional supplementary Fig. S14. The value of 51% was a calculation error, and was corrected to 41%. Likewise, the decrease in the WTA/LTA ratio was corrected to 5 to 7-fold.

(7) 14. l. 385-391. Contrary to the statement in the text, the zwitterionic TA will have associated counterions that result in net neutrality. It will just have both -ve and +ve counterions in equal amounts (dependent on their valency), which doesn't matter if it is doing the job of balancing osmolarity (rather than charge).

Thank you for pointing out this point. The paragraph has been corrected in V2.

**Reviewer #2 (Public review):**
The Gram-positive cell wall contains for a large part of TAs, and is essential for most bacteria. However, TA biosynthesis and regulation is highly understudied because of the difficulties in working with these molecules. This study closes some of our important knowledge gaps related to this and provides new and improved methods to study TAs. It also shows an interesting role for TAs in maintaining a 'periplasmic space' in Gram positives. Overall, this is an important piece of work. It would have been more satisfying if the possible causal link between TAs and periplasmic space would have been more deeply investigated with complemented mutants and CEMOVIS. For the moment, there is clearly something happening but it is not clear if this only happens in TA mutants or also in strains with capsules/without capsules and in PG mutants, or in lafB (essential for production of another glycolipid) mutants. Finally, some very strong statements are made suggesting several papers in the literature are incorrect, without actually providing any substantiation/evidence supporting these claims. Nevertheless, I support the publication of this work as it pioneers some new methods that will definitively move the field forward.
**Recommendations for the authors:**

**Reviewer #1 (Recommendations for the authors):**
(1) l. 55 It is stated that TA are generally not essential. This needs to be introduced in a little more detail as in several species they are collectively. Need some more references here to give context.

We have expended the paragraph and added a selection of references in V2.

(2) l. 63 and Fig. 1A. Is the model based on the images from this paper? Is the periplasm as thick as the peptidoglycan layer? Would you not expect the density of WTA to be the same throughout the wall, rather than less inside? Do the authors think that the TA are present as rods in the cell envelope and because of this the periplasm looks a little like a bilayer, is this so? Is the relative thickness of the layers based on the data in the paper (Table 1)?

The model proposed in Fig. 1A is not based on our data. It is a representation of the model proposed by Harold Erickson, and the appropriate reference has been added to the figure legend in V2. We do not speculate on the relative density of WTA inside the peptidoglycan layer, at the surface or in the periplasm. The only constraint from the model is that the density of WTA in the periplasm should be sufficient for self-exclusion and allow the brush polymer theory to apply. The legend has been amended in V2.

We indeed think that the bilayer appearance of the periplasmic space in the wild type strain, and the single layer periplasmic space in the ∆*tacL* and ∆*lytR* support the Erickson’s model. Although the model was drawn arbitrarily, it turns out that the relative thickness of the peptidoglycan and periplasmic scale is in rough agreement with the measurements reported in Table 1.

(3) Fig. 2. It is hard to orient oneself to see the layers. The use of the term periplasmic space (l. 132) and throughout is probably not wise as it is not a space.

We prefer to retain this nomenclature since the term periplasmic space has been used in all the cell envelope CEMOVIS publications and is at the core of Erickson’s hypothesis about these observations and teichoic acids.

(4) L. 147. This is not referring to Fig. S2A-B as suggested but Fig. S3A-B.

This has been corrected.

(5) l. 148. How do you know the densities observed are due to PG or certainly PG alone? Perhaps it is better to call this the cell wall.

Yes. Cell wall is a better nomenclature and the text and Table 1 have been corrected in V2, in accordance with Fig. 2.

(6) l. 165. It is also worth noting that peripheral cell wall synthesis also happens at the same site so this may well not be just division.

Yes. We have replaced “division site” by “mid-cell” in V2.

(7) l. 214 What is the debris? If PG digestion has been successful then there will be marginal debris. Is this pellet translucent (like membranes)? If you use fluorescently labelled PG in the preparation has it all disappeared, as would be expected by fully digested and solubilised material?

In traditional protocols of bacterial membrane preparation, a low-speed centrifugation is first performed to discard “debris” that to our knowledge have not been well characterized but are thought to consist of unbroken cells and large fragments of cell wall. After enzymatic degradation of the pneumococcal cell wall, the low-speed pellet is not translucent as in typical membrane pellets after ultracentrifugation, but is rather loose, unlike a dense pellet of unbroken cells. A description of the pellet appearance was added in V2.

It is a good idea to check if some labeled PG is also pelleted at low-speed after digestion. In a double labeling experiment using azido-choline and a novel unpublished metabolic probe of the PG, we found that the PG was fully digested and labeled fragments migrated as a couple of fuzzy bands likely corresponding to different labeled peptides. These species were not pelleted at low speed.

(8) l. 219. Can you give a reference to certify that the low mobility material is WTA? Why does it migrate differently than LTA? Or is the PG digestion not efficient?

WTA released from sacculi by alkaline lysis were found to migrate as a smear at the top of native gels revealed by alcian-blue silver staining, which is incompatible with SDS (Flores-Kim, 2019, 2022). The references have be added in V2. It could be argued in this case that the smearing was due to partial degradation of the WTA by the alkaline treatment.

Bui et al. (2012) reported the preparation of WTA by enzymatic digestion of sacculi, but the resulting WTA were without muropeptide, presumably due to a step of boiling at pH 5 used to deactivate the enzymes.

To our knowledge, this is the first report of pneumococcal WTA prepared by digestion of sacculi and analyzed by SDS-PAGE. Since the migration of WTA in native and SDS-PAGE is similar, we hypothesize that they do not interact significantly with the dodecyl sulphate, in contrast to the LTA, which bear a lipidic moiety. The fuzziness of the WTA migration pattern may also result from the greater heterogeneity due to the attached muropeptide, such as different lengths (di-, tetra-saccharide…), different peptides despite the action of LytA (tri-, tetra-peptide…), different O-acetylation status, etc.

(9) L. 226-227, Fig S8. Presumably several of the major bands on the Coomassie stained gel are the lysozyme, mutanolysin, recombinant LytA, DNase and RNase used to digest the cell wall etc.? Can the sizes of these proteins be marked on the gel. Do any of them come down with the material at low-speed centrifugation?

We have provided a gel showing the different enzymes individually and mixed (new Fig. S9G). While performing several experiments of this type, we found that the mutanolysin might be contaminated with proteases. The enzymes do not appear to sediment at low speed.

(10) Fig. S9B. It is difficult to interpret what is in the image as there appear to be 2 populations of material (grey and sometimes more raised). Does the 20,000 g material look the same?

Fig. S10B is a 20,000 × *g* pellet. We agree that there appears to be two types of membrane vesicles, but we do not know their nature.

(11) l. 277 and Fig. 5A. Why is it "remarkable" that there are apparently more longer LTA molecules as the cell reach stationary phase?

This is the first time that a change of TA length is documented. Such a change could conceivably have consequences in the binding and activity of CBPs and the physiology of the cell envelope in general. These questions should be adressed in future studies.

(12) l. 280. How do you know which is the 6-repeat unit?

It is an assumption based on previous analyses by Gisch et al.(J Biol Chem 2013, 288(22):15654-67. doi: 10.1074/jbc.M112.446963). The reference was added.

(13) Fig. 5A and C. Panel C, the cells were grown in a different medium and so are not comparable to Panel A. Why is Fig. S12B not substituted for 5B? Presumably these are exponential phase cells.

We have interverted the Fig. S13B and 5C in V2, as suggested, and changed the text and legends accordingly.

**Reviewer #2 (Recommendations for the authors):**
L30: vitreous sections?

Corrected in V2.

L32: as their main universal function  as a universal function. To show it's the main universal function, you will need to look at this across various bacterial species.

Changed to “possible universal function” in V2.

L35: enabled the titration the actual  titration of the actual?

Corrected in V2.

L34: consider breaking up this very long sentence.

Done in V2.

L37: may compensate the absence may compensate for the absence.

Corrected in V2.

L45: Using metabolic labeling and electrophoresis showed  Metabolic labeling and...

Corrected in V2.

L46: This finding casts doubts on previous results, since most LTA were likely unknowingly discarded in these studies. This needs to be rephrased and is unnecessarily callous. While the current work casts doubts on any quantitative assessments of actual LTA levels measured in previous studies, it does not mean any qualitative assessments or conclusions drawn from these experiments are wrong. Better would be to say: These findings suggest that previously reported quantitative assessments of LTA levels are likely underestimating actual LTA levels, since much of the LTA would have been unknowingly discarded.

If the authors do think that actual conclusions are wrong in previous work, then they need to be more explicit and explain why they were wrong.

Yes indeed. The statement was toned down in V2.

L55: Although generally non-essential. I would remove or rephrase this statement. I don't think any TA mutant will survive out in the wild and will be essential under a certain condition. So perhaps not essential for growth under ideal conditions, but for the rest pretty essential.

The paragraph was amended by qualifying the essentiality to laboratory conditions and including selected references.

L95: Note that the prevailing model until reference 20 (Gibson and Veening) was that the TA is polymerized intracellularly (see e.g. Figure 2 of PMID: 22432701, DOI: 10.1089/mdr.2012.0026). This intracellular polymerisation model seemed unlikely according to Gibson and Veening ('As TarP is classified by PFAM as a Wzy-type polymerase with predicted active site outside the cell, we speculate that TarP and TarQ polymerize the TA extracellularly in contrast to previous reports.'), but there is no experimental evidence as far as this referee knows of either model being correct.

Despite the lack of experimental evidence, we think that Gibson and Veening are very likely correct, based on their argument, and also by analogy with the synthesis of other surface polysaccharides from undecaprenyl- or dolichol-linked precursors. It is unfortunate that Figure 2 of PMID: 22432701, DOI: 10.1089/mdr.2012.0026 was published in this way, since there was no evidence for a cytoplasmic polymerization, to our knowledge.

L97: It is commonly believed, although I'm not sure it has ever been shown, that the capsule is covalently attached at the same position on the PG as WTA. Therefore, there must be some sort of regulation/competition between capsule biosynthesis and WTA biosynthesis (see also ref. 21). The presence of the capsule might thus also influence the characteristics of the periplasmic space. Considering that by far most pneumococcal strains are encapsulated, the authors should discuss this and why a capsule mutant was used in this study and how translatable their study using a capsule mutant is to S. pneumoniae in general.

A paragraph was added in the Introduction of V2 to present the complication and a sentence was added at the end of the discussion to mention that this should be studied in the future.

L102: Ref 29 should probably be cited here as well?

Since in Ref 29 (Flores-Kim et al. 2019) there is a detectable amount of LTA (presumably precursors TA) in the ∆*tacL* stain, we prefer to cite only Hess et al. 2017 regarding the absence of LTA in the absence of TacL. However, we added in V2 a reference to Flores-Kim et al. 2019 in the following paragraph regarding the role of the LTA/WTA ratio.

L106: dependent on the presence of the phosphotransferase LytR (21).  dependent on the presence of the phosphotransferase LytR, whose expression is upregulated during competence (21).

Corrected in V2.

L119: I fail to see how the conclusions drawn by other groups (I assume the authors mean work from the Vollmer, Rudner, Bernhardt, Hammerschmidt, Havarstein, Veening groups?) are invalid if they compared WTA:LTA ratios between strains and conditions if they underestimated the LTA levels? Supposedly, the LTA levels were underestimated in all samples equally so the relative WTA/LTA ratio changes will qualitatively give the same outcome? I agree that these findings will allow for a reassessment of previous studies in which presumably too low LTA levels were reported, but I would not expect a difference in outcome when people compared WTA:LTA ratios between strains?

The sentence was rephrased in V2 to be neutral regarding previous work and rather emphasize future possibilities.

L131: Perhaps it would be good to highlight that such a conspicuous space has been noticed before by other EM methods (see e.g. Figs.4 and 5 or ref 19, or one of the most clear TEM S. pneumoniae images I have seen in Fig. 1F of Gallay et al, Nat. Micro 2021). However, always some sort of staining had previously been performed so it was never clear this was a real periplasmic space. CEMOVIS has this big advantage of being label free and imaging cells in their presumed native state.

Thanks for pointing out these beautiful data that we had overlooked. We have added a few sentences and references in the Discussion of V2.

L201: References are not numbered.

Corrected in V2.

L271/L892: Change section title. 'Evolution' can have multiple meanings. It would be more clear to write something like 'Increased TA chain length in stationary phase cells' or something like that.

Changed in V2.

L275: harvested

Corrected in V2.

L329: add, as suggested shown previously (I guess refs 24 and 29)

Reference to Hess et al. 2017 has been added in V2. A sentence and further references to Flores-Kim, 2019, 2022 and Wu et al. 2014 were added at the end of the discussion with respect to the LTA-like signal observed in these studies of ∆*tacL* strains.

L337: I think a concluding sentence is warranted here. These experiments demonstrate that membrane-bound TA precursors accumulate on the outside of the membrane, and are likely polymerized on the outside as well, in line with the model proposed in ref. 20.

From the point of view of formal logic, the accumulation of membrane-bound TA precursors on the outer face of the membrane does not prove that they were assembled there. They could still be polymerized inside and translocated immediately. However, since this is extremely unlikely for the reasons discussed by Gibson and Veening, we have added a mild conclusion sentence and the reference in V2.

L343: How accurate are these quantifications? Just by looking at the gel, it seems there is much less WTA in the lytR mutant than 50% of the wild type?

Yes, the 51% value was a calculation error. This was changed to 41%. Likewise, the decrease of the WTA amount relative to LTA was corrected to 5- to 7-fold.

Apart from the titration of TA in the WT strain, we haven’t yet carried out a careful quantification neither of TA nor of the LTA/WTA ratio in different strains and conditions, although we intend to do so in the near future using the method presented here.

However, to better substantiate our statement regarding the ∆*lytR* strain, we have quantified two experiments of growth in C-medium with azido-choline, and two experiments of pulse labeling in BHI medium. The results are presented in the additional supplementary Fig. S14.

L342: although WTA are less abundant and LTA appear to be longer (Fig. 6A). although WTA are less abundant and LTA appear to be longer (Fig. 6A), in line with a previous report showing that LytR the major enzyme mediating the final step in WTA formation (ref. 21). (or something like that). Perhaps better is to start this paragraph differently. For instance: Previous work showed that LytR is the major enzyme mediating the final step in WTA formation (ref. 21). As shown in Fig. 6A, the proportion of WTA significantly decreased in the lytR mutant. However, there was still significant WTA present indicating that perhaps another LCP protein can also produce WTA.

Changed in V2.

Of note, WTA levels would be a lot lower in encapsulated strains as used in Ref. 21 (assuming WTA and capsule compete for the same linkage on PG). So perhaps it would be hard to detect any residual WTA in a encapsulated lytR mutant?

Investigation of the relationship between TA and capsule incorporation or O-acetylation is definitely a future area of study using this method of TA monitoring.

L371: see my comments related to L131. Some TEM images clearly show the presence of a periplasmic space.

Comments and references have been added in V2.

L402: It would be really interesting to perform these experiments on a wild type encapsulated strain. Would these have much more LTA? (I understand you cannot do these experiments perhaps due to biosafety, but it might be interesting to discuss).

Yes. It would be interesting to compare the TA in D39 and D39 ∆*cps* strains. We have added this perspective at the end of the discussion in V2.

L418: ref lacks number

Corrected in V2.

L423: refs missing.

References added in V2.

L487: See my comments regarding L46. I do not see one valid point in the current paper why underestimating LTA levels would change any of the conclusions drawn in Ref. 21. I do not know the other papers cited well enough, but it seems highly unlikely that their conclusions would be wrong by systematically underestimating LTA levels. As far as I understand it, this current work basically confirms the major conclusions drawn by these 'doubtful' papers (that TacL makes LTA and LytR is the main WTA producer). As such, I find this sentence highly unfair without precisely specifying what the exact doubts are. Sure, this current paper now shows that probably people have discarded unknowingly LTA and therefore underestimated LTA levels, so any quantitative assessment of LTA levels are probably wrong. That is one thing. But to say this casts doubts on these studies is very serious and unfair (unless the authors provide good arguments to support these serious claims).

Yes indeed. The sentence was rephrased to be strictly factual in V2.

Table 2: I assume these strains are delta cps? Would be relevant to list this genotype.

The Table 2 was completed in V2.

The authors should comment on why the mutants have not been complemented, especially for lytR as it's the last gene in a complex operon. It would be great to see WTA levels being restored by ectopic expression of LytR.

Yes. We think this could be part of an in-depth study of the attachment of WTA, together with the investigation of the other LCP phosphotransferases.